# Persistent coding of outcome-predictive cue features in the rat nucleus accumbens

**Jimmie M Gmaz, James E Carmichael, Matthijs AA van der Meer***

Department of Psychological and Brain Sciences, Dartmouth College, Hanover, United States

**Abstract** The nucleus accumbens (NAc) is important for learning from feedback, and for biasing and invigorating behaviour in response to cues that predict motivationally relevant outcomes. NAc encodes outcome-related cue features such as the magnitude and identity of reward. However, little is known about how features of cues themselves are encoded. We designed a decision making task where rats learned multiple sets of outcome-predictive cues, and recorded single-unit activity in the NAc during performance. We found that coding of cue identity and location occurred alongside coding of expected outcome. Furthermore, this coding persisted both during a delay period, after the rat made a decision and was waiting for an outcome, and after the outcome was revealed. Encoding of cue features in the NAc may enable contextual modulation of on-going behaviour, and provide an eligibility trace of outcome-predictive stimuli for updating stimulus-outcome associations to inform future behaviour.

DOI: https://doi.org/10.7554/eLife.37275.001

## Introduction

Theories of nucleus accumbens (NAc) function generally agree that this brain structure contributes to motivated behaviour, with some emphasizing a role in learning from reward prediction errors (RPEs) (*Averbeck and Costa, 2017*; *Joel et al., 2002*; *Khamassi and Humphries, 2012*; *Lee et al., 2012*; *Maia, 2009*; *Schultz, 2016*; see also the addiction literature on effects of drug rewards; *Carelli, 2010*; *Hyman et al., 2006*; *Kalivas and Volkow, 2005*), and others a role in the modulation of on-going behaviour through stimuli associated with motivationally relevant outcomes (invigorating, directing; *Floresco, 2015*; *Nicola, 2010*; *Salamone and Correa, 2012*). These proposals echo similar ideas on the functions of the neuromodulator dopamine (*Berridge, 2012*; *Maia, 2009*; *Salamone and Correa, 2012*; *Schultz, 2016*), with which the NAc is tightly linked functionally as well as anatomically (*Cheer et al., 2007*; *du Hoffmann and Nicola, 2014*; *Ikemoto, 2007*; *Takahashi et al., 2016*).

Much of our understanding of NAc function comes from studies of how cues that predict motivationally relevant outcomes (e.g. reward) influence behaviour and neural activity in the NAc. Task designs that associate such cues with rewarding outcomes provide a convenient access point, eliciting conditioned responses such as sign-tracking and goal-tracking (*Hearst and Jenkins, 1974*; *Robinson and Flagel, 2009*), Pavlovian-instrumental transfer (*Estes, 1943*; *Rescorla and Solomon, 1967*) and enhanced response vigor (*Nicola, 2010*; *Niv et al., 2007*), which tend to be affected by NAc manipulations (*Chang et al., 2012*; *Corbit and Balleine, 2011*; *Flagel et al., 2011*; although not always straightforwardly; *Chang and Holland, 2013*; *Giertler et al., 2004*). Similarly, analysis of RPEs typically proceeds by establishing an association between a cue and subsequent reward, with NAc responses transferring from outcome to the cue with learning (*Day et al., 2007*; *Roitman et al., 2005*; *Schultz et al., 1997*; *Setlow et al., 2003*).

Surprisingly, although substantial work has been done on the coding of outcomes predicted by such cues (*Atallah et al., 2014*; *Bissonette et al., 2013*; *Cooch et al., 2015*; *Cromwell and Schultz,*

***For correspondence:**
mvdm@dartmouth.edu

**Competing interests:** The authors declare that no competing interests exist.

*2003*; *Day et al., 2006*; *Goldstein et al., 2012*; *Hassani et al., 2001*; *Hollerman et al., 1998*; *Lansink et al., 2012*; *McGinty et al., 2013*; *Nicola et al., 2004*; *Roesch et al., 2009*; *Roitman et al., 2005*; *Saddoris et al., 2011*; *Setlow et al., 2003*; *Sugam et al., 2014*; *Schultz et al., 1992*; *West and Carelli, 2016*), much less is known about how outcome-predictive cues themselves are encoded in the NAc (but see; *Sleezer et al., 2016*). This is an important issue for at least two reasons. First, in reinforcement learning, motivationally relevant outcomes are typically temporally delayed relative to the cues that predict them. In order to solve the problem of assigning credit (or blame) across such temporal gaps, some trace of preceding activity needs to be maintained (*Lee et al., 2012*; *Sutton and Barto, 1998*). For example, if you become ill after eating food X in restaurant A, depending on if you remember the identity of the restaurant or the food at the time of illness, you may learn to avoid all restaurants, restaurant A only, food X only, or the specific pairing of X-in-A. Therefore, a complete understanding of what is learned following feedback requires understanding what 'trace' is maintained. Since NAc is a primary target of dopamine signals interpretable as RPEs, and NAc lesions impair RPEs related to timing, its activity trace will help determine what can be learned when RPEs arrive (*Hamid et al., 2016*; *Hart et al., 2014*; *Ikemoto, 2007*; *McDannald et al., 2011*; *Takahashi et al., 2016*). Similarly, in a neuroeconomic framework, NAc is thought to represent a domain-general subjective value signal for different offers (*Peters and Büchel, 2009*; *Levy and Glimcher, 2012*; *Bartra et al., 2013*; *Sescousse et al., 2015*); having a representation of the offer itself alongside this value signal would provide a potential neural substrate for updating offer value.

Second, for on-going behaviour, the relevance of cues typically depends on 'context'. In experimental settings, context may include the identity of a preceding cue, spatial or configural arrangements (*Bouton, 1993*; *Holland, 1992*; *Honey et al., 2014*), and unsignaled rule changes, as occurs in set shifting and other cognitive control tasks (*Cohen and Servan-Schreiber, 1992*; *Floresco et al., 2006*; *Grant and Berg, 1948*; *Sleezer et al., 2016*). In such situations, the question arises how selective, context-dependent processing of outcome-predictive cues is implemented. For instance, is there a 'gate' prior to NAc such that only currently relevant cues are encoded in NAc, or are all cues represented in NAc but their current values dynamically updated (*FitzGerald et al., 2014*; *Goto and Grace, 2008*; *Sleezer et al., 2016*)? Representation of cue identity would allow for context-dependent mapping of outcomes predicted by specific cues.

Thus, both from a learning and a flexible performance perspective, it is of interest to determine how cue identity is represented in the brain, with NAc of particular interest given its anatomical and functional position at the centre of motivational systems. We sought to determine whether cue identity is represented in the NAc, if cue identity is represented alongside other motivationally relevant variables, such as cue outcome, and if these representations are maintained after a behavioural decision has been made (see *Figure 1* for a schematic representation of the specific hypotheses tested). To address these questions, we recorded the activity of NAc units as rats performed a task in which multiple, distinct sets of cues predicted the same outcome.

## Results

### Behaviour

Rats were trained to discriminate between cues signalling the availability and absence of reward on a square track with four identical arms for two distinct set of cues (*Figure 2A*). During each session, rats were presented sequentially with two behavioural blocks containing cues from different sensory modalities, a light and a sound block, with each block containing a cue that signalled the availability of reward (reward-available), and a cue that signalled the absence of reward (reward-unavailable). To maximize reward receipt, rats should approach reward sites on reward-available trials, and skip reward sites on reward-unavailable trials (see *Figure 2B* for an example learning curve). All four rats learned to discriminate between the reward-available and reward-unavailable cues for both the light and sound blocks as determined by reaching significance ($p < .05$) on a daily chi-square test comparing approach behaviour for reward-available and reward-unavailable cues for each block, for at least three consecutive days (range for time to criterion: 22–57 days). Maintenance of behavioural performance during recording sessions was assessed using linear mixed effects models for proportion of trials where the rat approached the receptacle. Analyses revealed that the likelihood of a rat to

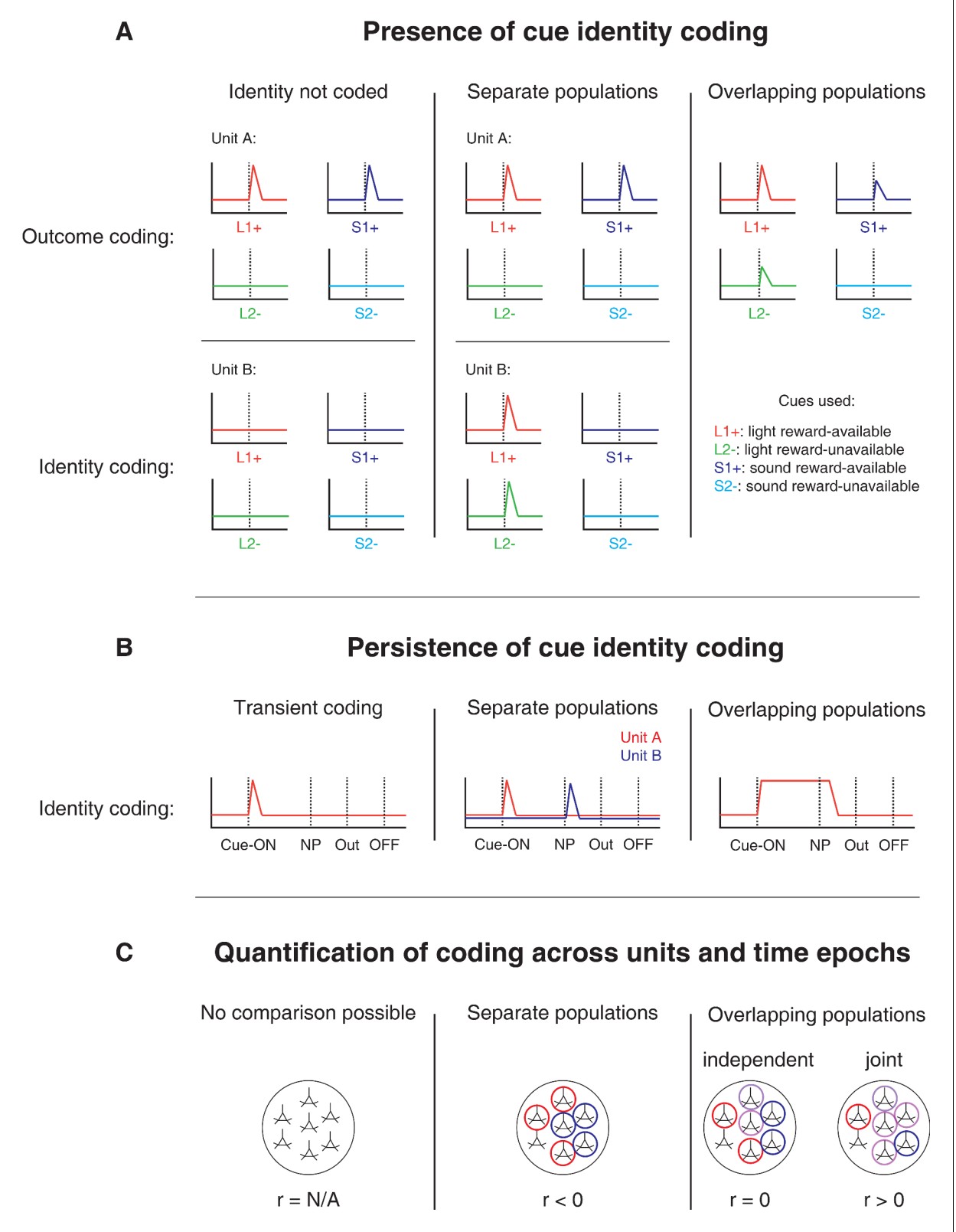

**Figure 1.** Schematic of hypothetical coding scenarios for cue feature coding employed by single units in the NAc across different cue features (**A**) and phases of a trial (**B**). (**A**) Schematic peri-event time histograms (PETHs) illustrating putative responses to different cues under different hypotheses of how cue identity (light, sound; L, S) and outcome (reward-available, reward-unavailable; +, -) are coded. Left panel: Coding of identity is absent in the NAc. Top: Unit A encodes a motivationally relevant variable, such as expected outcome, similarly across other cue features, such as identity or physical

*Figure 1 continued on next page*

*Figure 1 continued*

location. Hypothetical plot is firing rate across time. L1+ (red) signifies a reward-available light cue, S1+ (navy blue) a reward-available sound cue, L2- (green) a reward-unavailable light cue, S2- (light blue) a reward-unavailable sound cue. Dashed line indicates onset of cue. Bottom: No units within the NAc discriminate their firing according to cue identity. Middle panel: Coding of identity occurs in a separate population of units from coding of other cue features such as expected outcome or physical location. Top: Same as left panel, with unit A discriminating between reward-available and reward-unavailable cues. Bottom: Unit B discriminates firing across stimulus modalities, depicted here as firing to light cues but not sound cues. Note lack of coding overlap in both units. Right panel: Coding of identity occurs in an overlapping population of cells with coding of other motivationally relevant variables. Hypothetical example demonstrating a unit that responds to reward-available cues, but firing rate is also modulated by the stimulus modality of the cue, firing most for the reward-available light cue. (**B**) Schematic PETHs illustrating potential ways in which identity coding may persist over time. Left panel: Cue-onset triggers a transient response to a unit that codes for cue identity. Dashed lines indicate time of a behavioural or environmental event. 'Cue-ON' signifies cue-onset, 'NP' signifies nosepoke at a reward receptacle, 'Out' signifies when the outcome is revealed, 'OFF' signifies cue-offset. Middle and right panel: Identity coding persists at other time points, shown here during a nosepoke hold period until outcome is revealed. Coding can either be maintained by a sequence of units (middle panel) or by the same unit as during cue-onset (right panel). (**C**) Schematic pool of NAc units, illustrating different analysis outcomes that discriminate between hypotheses. *R* values represent the correlation between sets of recoded regression coefficients (see text for analysis details). Left panel: Cue identity is not coded (A: left panel), or is only transiently represented in response to the cue (B: left panel). Middle panel: Negative correlation (r < 0) suggests that identity and outcome coding are represented by separate populations of units (A: middle panel), or identity coding is represented by distinct units across different points in a trial (B: middle panel). Red circles represents coding for one cue feature or point in time, blue circles for the other cue feature or point in time. Right panel: Identity and outcome coding (A: right panel), or identity coding at cue-onset and nosepoke (B: right panel) are represented by overlapping populations of units, shown here by the purple circles. The absence of a correlation (r = 0) suggests that the overlap of identity and outcome coding, or identity coding at cue-onset and nosepoke, is expected by chance and that the two cue features, or points in time, are coded by overlapping but independent populations from one another. A positive correlation (r > 0) implies a higher overlap than expected by chance, suggesting coding by a joint population. Note: The same logic applies to other aspects of the environment when the cue is presented, such as the physical location of the cue, as well as other time epochs within the task, such as when the animal receives feedback about an approach.

DOI: https://doi.org/10.7554/eLife.37275.002

make an approach was influenced by whether a reward-available or reward-unavailable cue was presented, but was not significantly modulated by whether the rat was presented with a light or sound cue (Percentage approached: light reward-available = 97%; light reward-unavailable = 34%; sound reward-available = 91%; sound reward-unavailable = 35%; cue identity p = .115; cue outcome p < .001; *Figure 2C*). Additional analyses separated each block into two halves to assess possible within session learning. Adding block half into the model did not improve prediction of behavioural performance (p = .86), arguing against within-session learning. Thus, rats successfully discriminated the cues according to whether or not they signalled the availability of reward at the reward receptacle.

## NAc encodes behaviourally relevant and irrelevant cue features

We sought to address which parameters of our task were encoded by NAc activity, specifically whether the NAc encodes aspects of motivationally relevant cues not directly tied to reward, such as the identity and location of the cue, and whether this coding is accomplished by separate or overlapping populations (*Figure 1A*). We recorded a total of 443 units with > 200 spikes in the NAc from 4 rats over 57 sessions (*Table 1*; range: 12 – 18 sessions per rat) while they performed a cue discrimination task (*Figure 2A*). Units that exhibited a drift in firing rate over the course of either block, as measured by a Mann-Whitney U test comparing firing rates for the first and second half of trials within a block, were excluded from further analysis, leaving 344 units for further analysis. The activity of 133 (39%) of these 344 units were modulated by the cue, as determined by comparing 1 s pre- and post-cue activity with a Wilcoxon signed-rank test, with more showing a decrease in firing (n = 103) than an increase (n = 30) around the time of cue-onset (*Table 1*). Within this group, 24 were classified as putative fast spiking interneurons (FSIs), while 109 were classified as putative medium spiny neurons (MSNs). Upon visual inspection, we observed several patterns of firing activity, including units that discriminated firing upon cue-onset across various cue conditions, showed sustained differences in firing across cue conditions, had transient responses to the cue, showed a ramping of activity starting at cue-onset, and showed elevated activity immediately preceding cue-onset (*Figure 3*, *Figure 3—figure supplement 1* and *Figure 3—figure supplement 2*).

To characterize more formally whether these cue-modulated responses were influenced by various aspects of the task, we fit a sliding window generalized linear model (GLM) to the firing rate of

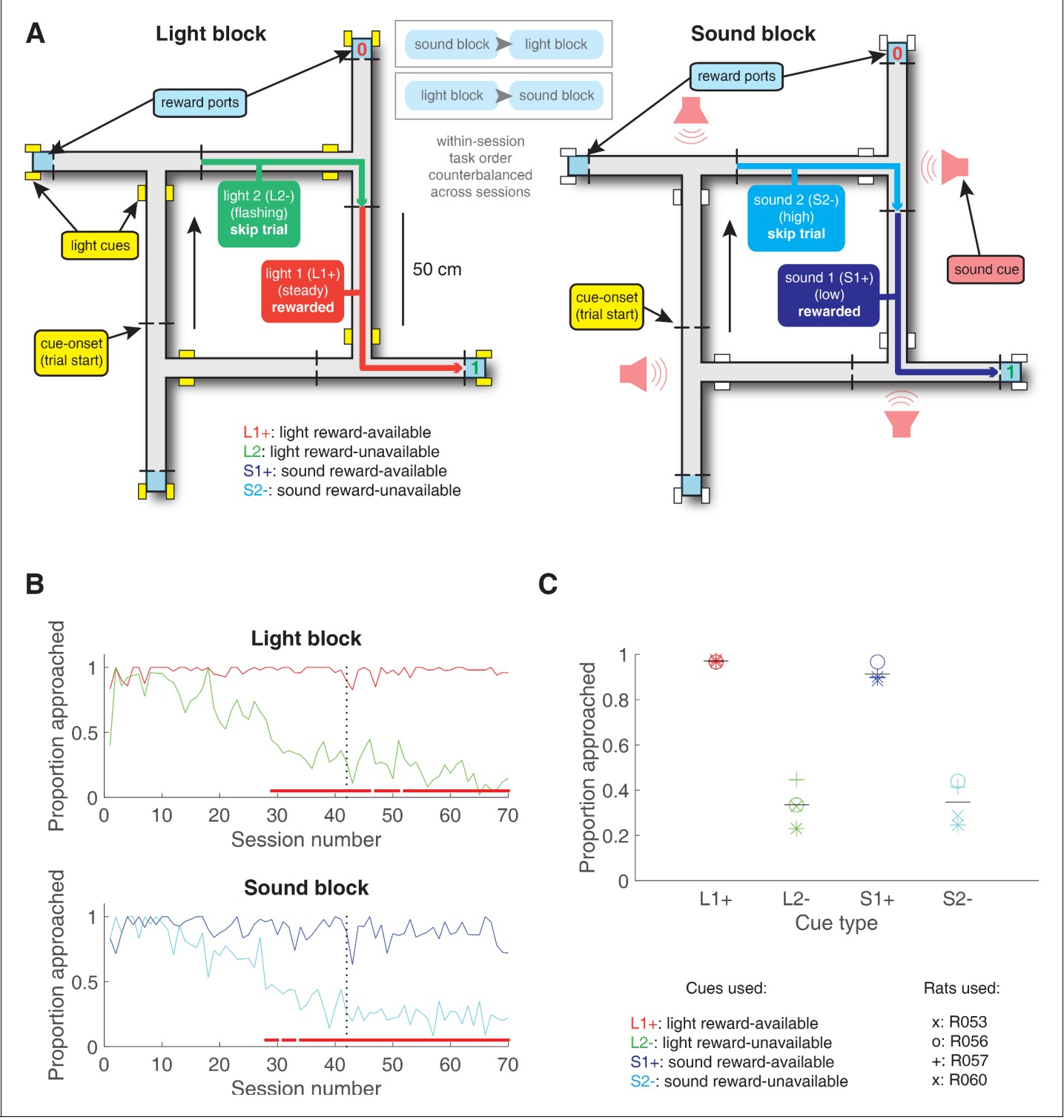

**Figure 2.** Schematic and performance of the behavioural task. (**A**) Apparatus was a square track consisting of multiple identical T-choice points. At each choice point, the availability of 12% sucrose reward at the nearest reward receptacle (light blue fill) was signalled by one of four possible cues, presented when the rat initiated a trial by crossing a photobeam on the track (dashed lines). Photobeams at the ends of the arms by the receptacles registered nosepokes. Arrows outside of track indicate correct running direction. Left: light block showing an example trajectory for a correct reward-available (approach trial; red) and reward-unavailable (skip trial; green) trial. Rectangular boxes with yellow fill indicate location of LEDs used for light cues. Right: sound block with a correct reward-available (approach trial; navy blue) and reward-unavailable (skip trial; light blue) trial. Speakers for sound cues were placed underneath the choice points, indicated by magenta speaker icons. Ordering of the light and sound blocks was counterbalanced

*Figure 2 continued on next page*

*Figure 2 continued*

across sessions. Reward-available and reward-unavailable cues were presented pseudo-randomly, such that not more than two of the same type of cue could be presented in a row. Location of the cue on the track was irrelevant for behaviour, all cue locations contained an equal amount of reward-available and reward-unavailable trials. (B-C) Performance on the behavioural task. B. Example learning curves across sessions from a single subject (R060) showing the proportion of trials approached for reward-available (red line for light block, navy blue line for sound block) and reward-unavailable trials (green line for light block, light blue line for sound block) for light (top) and sound (bottom) blocks. Fully correct performance corresponds to an approach proportion of 1 for reward-available trials and 0 for reward-unavailable trials. Rats initially approach on both reward-available and reward-unavailable trials, and learn with experience to skip reward-unavailable trials. Red bars indicate days in which a rat statistically discriminated between reward-available and reward-unavailable cues, determined by a chi square test. Dashed line indicates time of electrode implant surgery. (C) Proportion of trials approached for each cue, averaged across all recording sessions and shown for each rat. Different columns indicate the different cues (reward-available (red) and reward-unavailable (green) light cues, reward-available (navy blue) and reward-unavailable (light blue) sound cues). Different symbols correspond to individual subjects; horizontal black line shows the mean. All rats learned to discriminate between reward-available and reward-unavailable cues, as indicated by the clear difference of proportion approached between reward-available (~90% approached) and reward-unavailable cues (~30% approached), for both blocks (see Results for statistics).

DOI: https://doi.org/10.7554/eLife.37275.003

each cue-modulated unit surrounding cue-onset, using a forward selection stepwise procedure for variable selection, a bin size of 500 ms for firing rate and a step size of 100 ms for the sliding window. Fitting GLMs to all trials within a session revealed that a variety of task parameters accounted

**Table 1.** Overview of recorded NAc units and their relationship to task variables at various time epochs.
Percentage is relative to the number of cue-modulated units (n = 133).

| Task parameter | Total | ↑ MSN | ↓ MSN | ↑ FSI | ↓ FSI |
|---|---|---|---|---|---|
| All units | 443 | 155 | 216 | 27 | 45 |
| *Rat ID* | | | | | |
| R053 | 145 | 51 | 79 | 4 | 11 |
| R056 | 70 | 12 | 13 | 17 | 28 |
| R057 | 136 | 55 | 75 | 3 | 3 |
| R060 | 92 | 37 | 49 | 3 | 3 |
| Analysed units | 344 | 117 | 175 | 18 | 34 |
| Cue modulated units | 133 | 24 | 85 | 6 | 18 |
| *GLM aligned to cue-onset* | | | | | |
| Cue identity | 42 (32%) | 9 (38%) | 25 (29%) | 0 (-) | 8 (44%) |
| Cue location | 55 (41%) | 11 (46%) | 33 (39%) | 3 (50%) | 8 (44%) |
| Cue outcome | 26 (20%) | 5 (21%) | 15 (18%) | 1 (17%) | 5 (28%) |
| Approach behaviour | 32 (24%) | 8 (33%) | 19 (22%) | 2 (33%) | 3 (17%) |
| Trial length | 22 (17%) | 5 (21%) | 14 (16%) | 0 (-) | 3 (17%) |
| Trial number | 42 (32%) | 11 (46%) | 20 (24%) | 1 (17%) | 10 (56%) |
| Trial history | 8 (6%) | 1 (4%) | 5 (6%) | 0 (-) | 1 (6%) |
| *GLM aligned to nosepoke* | | | | | |
| Cue identity | 28 (21%) | 3 (13%) | 17 (20%) | 2 (33%) | 6 (33%) |
| Cue location | 30 (23%) | 2 (8%) | 21 (25%) | 2 (33%) | 5 (28%) |
| Cue outcome | 23 (17%) | 2 (8%) | 14 (16%) | 1 (17%) | 6 (33%) |
| *GLM aligned to outcome* | | | | | |
| Cue identity | 25 (19%) | 4 (17%) | 15 (18%) | 2 (33%) | 4 (22%) |
| Cue location | 31 (23%) | 5 (21%) | 23 (27%) | 0 (-) | 3 (17%) |
| Cue outcome | 34 (26%) | 6 (25%) | 15 (18%) | 4 (67%) | 9 (50%) |

DOI: https://doi.org/10.7554/eLife.37275.007

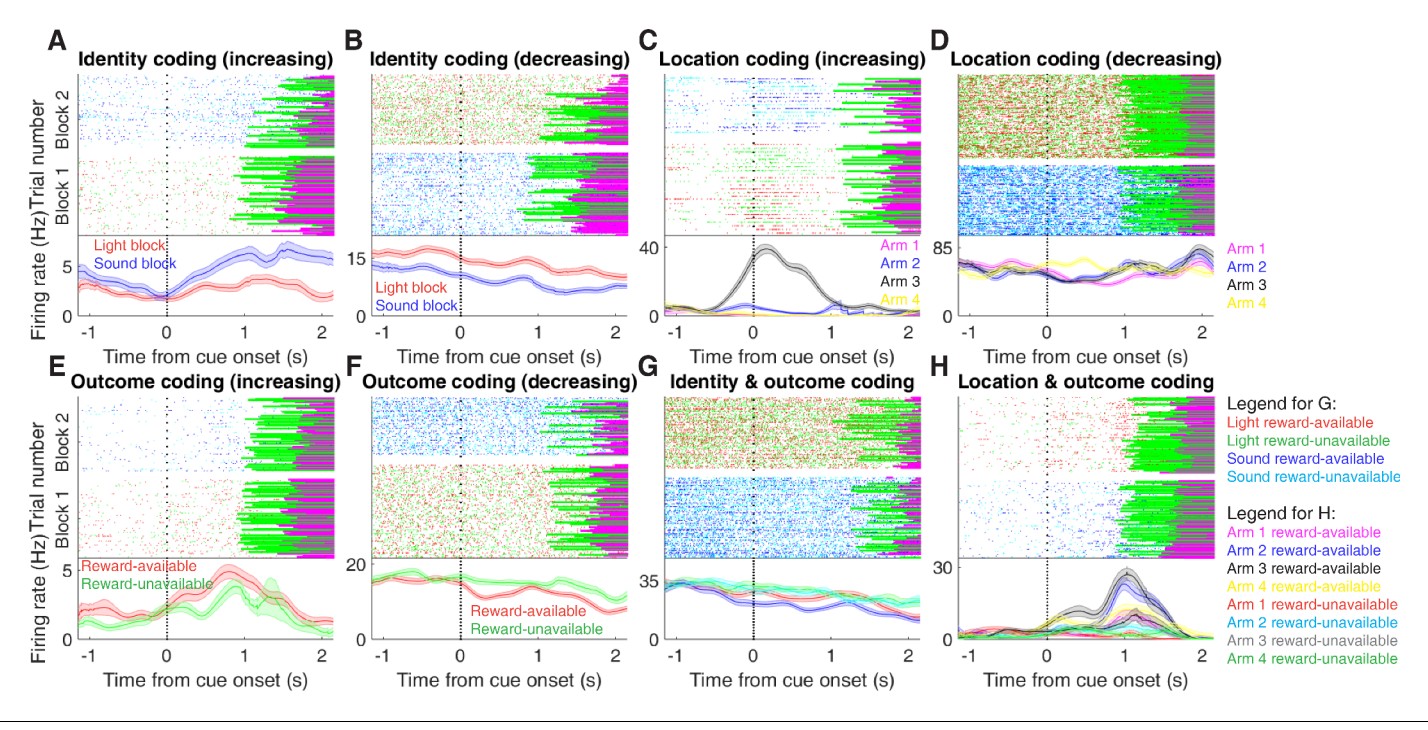

**Figure 3.** Examples of cue-modulated NAc units influenced by different task parameters. (**A**) Example of a cue-modulated NAc unit that showed an increase in firing following the cue, and exhibited identity coding. Top: raster plot showing the spiking activity across all trials aligned to cue-onset. Spikes across trials are colour-coded according to cue type (red: reward-available light; green: reward-unavailable light; navy blue: reward-available sound; light blue: reward-unavailable sound). Green and magenta bars indicate trial termination when a rat initiated the next trial or made a nosepoke, respectively. White space halfway up the raster plot indicates switching from one block to the next. Dashed line indicates cue-onset. Bottom: PETHs showing the average smoothed firing rate for the unit for trials during light (red) and sound (blue) blocks, aligned to cue-onset. Lightly shaded area indicates standard error of the mean. Note this unit showed a larger increase in firing to sound cues. (**B**) An example of a unit that was responsive to cue identity as in A, but for a unit that showed a decrease in firing to the cue. Note the sustained higher firing rate during the light block. (**C-D**) Cue-modulated units that exhibited location coding. Each colour in the PETHs represents average firing response for a different cue location. (**C**) The firing rate of this unit only changed on arm 3 of the task. (**D**) Firing rate decreased for this unit on all arms but arm 4. (**E-F**) Cue-modulated units that exhibited outcome coding, with the PETHs comparing reward-available (red) and reward-unavailable (green) trials. (**E**) This unit showed a slightly higher response during presentation of reward-available cues. (**F**) This unit showed a dip in firing when presented with reward-available cues. (**G-H**) Examples of cue-modulated units that encoded multiple cue features. (**G**) This unit showed both identity and outcome coding. (**H**) An example of a unit that coded for both identity and location.

DOI: https://doi.org/10.7554/eLife.37275.004

The following figure supplements are available for figure 3:

**Figure supplement 1.** Expanded examples of cue-modulated NAc units influenced by different task parameters for *Figure 3A–D*, showing firing rate breakdown by: cue type (top PETH), cue identity (top-middle PETH), cue location (bottom-middle PETH), and cue outcome (bottom PETH).
DOI: https://doi.org/10.7554/eLife.37275.005

**Figure supplement 2.** Expanded examples of cue-modulated NAc units influenced by different task parameters for *Figure 3E–H*, showing firing rate breakdown by: cue type (top PETH), cue identity (top-middle PETH), cue location (bottom-middle PETH), and cue outcome (bottom PETH).
DOI: https://doi.org/10.7554/eLife.37275.006

for a significant portion of firing rate variance in NAc cue-modulated units (*Figure 4A*, *Figure 4—figure supplement 1* and *Figure 4—figure supplement 2*, *Table 1*). Notably, a significant proportion of units discriminated between the light and sound block (*identity coding*: ~32% of cue-modulated units, accounting for ~5% of firing rate variance) or the arms of the apparatus (*location coding*: ~41% of cue-modulated units, accounting for ~4% of firing rate variance) throughout the entire window surrounding cue-onset. Additionally, a substantial proportion of units discriminating between the common portion of reward-available and reward-unavailable trials (*outcome coding*: ~20% of cue-modulated units, accounting for ~4% of firing rate variance) was not observed until after the onset of the cue (z-score > 1.96 when comparing observed proportion of units to a shuffled distribution obtained when shuffling the firing rates of each unit across trials before running the GLM).

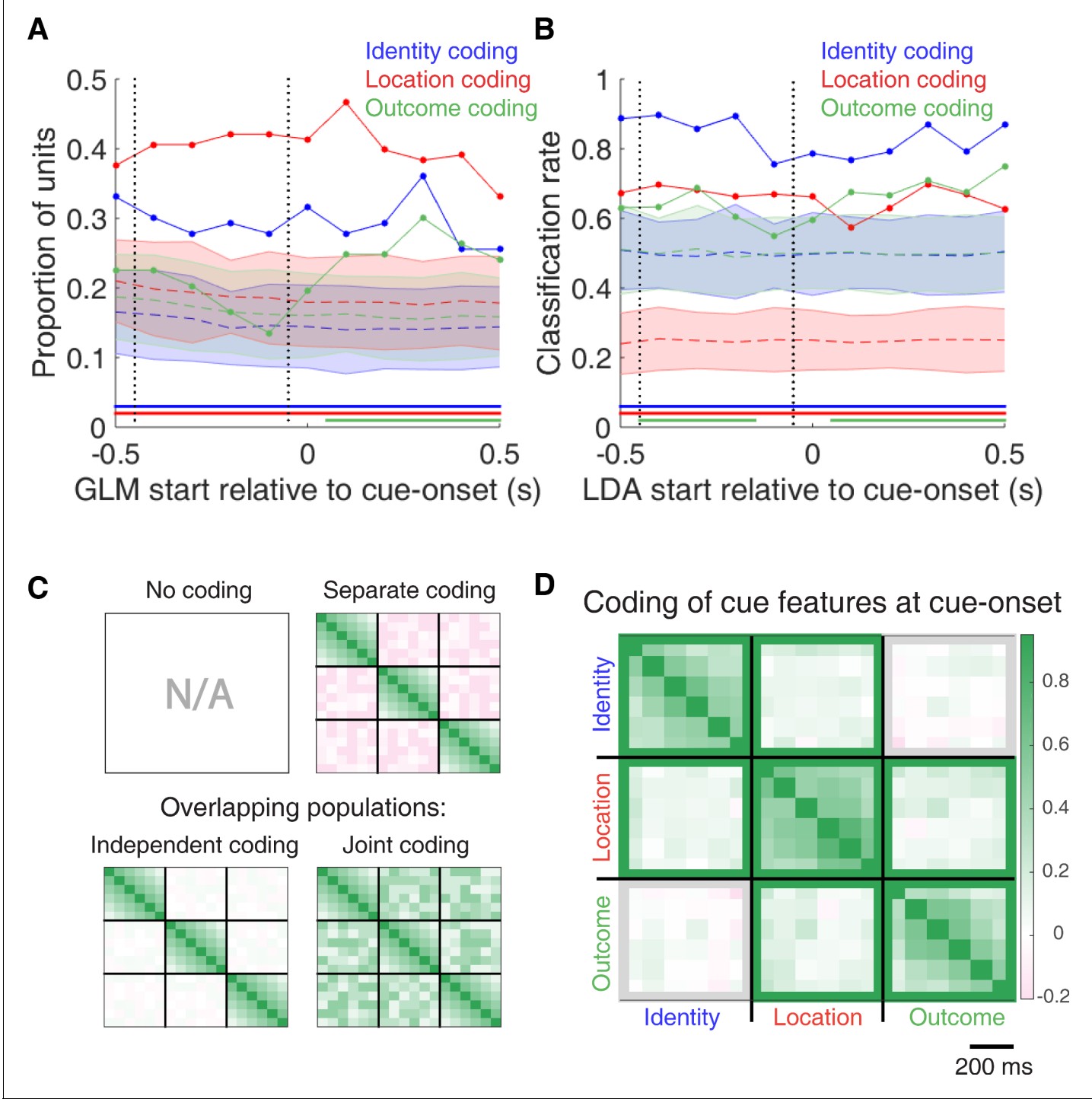

**Figure 4.** Summary of influence of cue features on cue-modulated NAc units at time points surrounding cue-onset. (**A**) Sliding window GLM (bin size: 500 ms; step size: 100 ms) demonstrating the proportion of cue-modulated units where cue identity (blue solid line), location (red solid line), and outcome (green solid line) significantly contributed to the model at various time epochs relative to cue-onset. Dashed coloured lines indicate the average of shuffling the firing rate order that went into the GLM 100 times. Error bars indicate 1.96 standard deviations from the shuffled mean. Solid lines at the bottom indicate when the proportion of units observed was greater than the shuffled distribution (z-score > 1.96). Points in between the two vertical dashed lines indicate bins where both pre- and post-cue-onset time periods were used in the GLM. (**B**) Sliding window LDA (bin size: 500 ms; step size: 100 ms) demonstrating the classification rate for cue identity (blue solid line), location (red solid line), and outcome (green solid line) using a pseudoensemble consisting of the 133 cue-modulated units. Dashed coloured lines indicate the average of shuffling the firing rate order that went into the cross-validated LDA 100 times. Solid lines at the bottom indicate when the classifier performance greater than the shuffled distribution (z-score >

*Figure 4 continued on next page*

*Figure 4 continued*

1.96). Points in between the two vertical dashed lines indicate bins where both pre- and post-cue-onset time periods were used in the classifier. (C-D) Correlation matrices testing the presence and overlap of cue feature coding at cue-onset. (C) Schematic outlining the possible outcomes for coding across cue features at cue-onset, generated by correlating the recoded beta coefficients from the GLMs and comparing to a shuffled distribution (see text for analysis details). Top left: coding is not present, therefore no comparison is possible. Top right: cue features are coded by separate populations of units. Displayed is a correlation matrix with each of the nine blocks representing correlations for two cue features across the post-cue-onset time bins from the sliding window GLM, with green representing positive correlations (r > 0), pink representing negative correlations (r < 0), and white representing no correlation (r = 0). X- and y-axis have the same axis labels, therefore the diagonal represents the correlation of a cue feature against itself at that particular time point (r = 1). Here the large amount of pink in the off-diagonal elements suggests that coding of cue features occur separately from one another. Bottom left: Coding of cue features occurs in overlapping but independent populations of units, shown here by the abundance of white and relative lack of green and pink in the off-diagonal elements. Bottom right: Coding of cue features occurs in a joint (correlated) overlapping population, shown here by the large amount of green in the off-diagonal elements. (D) Correlation matrix showing the correlation among cue identity, location, and outcome coding surrounding cue-onset. The window of GLMs used in each block is from cue-onset to the 500 ms window post-cue-onset, in 100 ms steps. Each individual value is for a sliding window GLM within that range, with the scale bar contextualizing step size. Colour bar displays relationship between correlation value and colour. Coloured square borders around each block indicate the result of a comparison of the mean correlation of a block to a shuffled distribution, with pink indicating separate populations (z-score < −1.96), grey indicating overlapping but independent populations, and green indicating joint overlapping populations (z-score > 1.96).

DOI: https://doi.org/10.7554/eLife.37275.008

The following figure supplements are available for figure 4:

**Figure supplement 1.** Summary of influence of various task parameters on cue-modulated NAc units at time points surrounding cue-onset.

DOI: https://doi.org/10.7554/eLife.37275.009

**Figure supplement 2.** Scatter plot depicting comparison of firing rates for cue-modulated units across light and sound blocks.

DOI: https://doi.org/10.7554/eLife.37275.010

Furthermore, our variable selection method ensured that the observed coding was not due to potential confounds from other task variables, such as behavioural response at the choice point (*approach behaviour*; left vs. right), variability in response vigor (*trial length*; see *McGinty et al., 2013*), drift due to the passage of time (*trial number*), and the pseudorandom nature of cue presentation (*trial history*). In addition to accounting for firing rate variance explained due to whether the rat turned left or right, we ran our cue-onset GLM using only approach trials, and found a similar proportion of outcome coding units (34 units; ~26% of cue-modulated units), providing further support that these units were coding the expected outcome of the cue. Taken together, these results from the GLMs suggest that the NAc encodes features of outcome-predictive cues in addition to expected outcome.

To assess what information may be encoded at the population level, we trained a classifier on a pseudoensemble of the 133 cue-modulated units (*Figure 4B*). Specifically, we used the firing rate of each unit for each trial as an observation, and different cue conditions as trial labels (e.g. light block, sound block). A linear discriminant analysis (LDA) classifier with 10-fold cross-validation could correctly predict a trial above chance levels for the identity and location of a cue across all time points surrounding cue-onset (z-score > 1.96 when comparing classification accuracy of data versus a shuffled distribution), whereas the ability to predict whether a trial was reward-available or reward-unavailable (outcome coding) was not significantly higher than the shuffled distribution for the time point containing 500 ms of pre-cue firing rate, and increased gradually as a trial progressed, providing evidence that cue information is also present at the pseudoensemble level.

To quantify the overlap of cue feature coding we correlated recoded beta coefficients from the GLMs, assigning a value of '1' if a cue feature was a significant predictor for that unit and '0' if not, and calculated a z-score comparing the mean of the obtained correlations to the mean and standard deviation of a shuffled distribution, generated by shuffling the unit ordering within an array (*Figures 1A,C* and *4C,D*). This revealed that identity was coded independently from outcome (mean r = .01, z-score = 0.81), and by a joint population with location (mean r = .10, z-score = 6.61), while location and outcome were coded by a joint population of units (mean r = .12, z-score = 8.07). Together, these findings show that various cue features are represented in the NAc at both the single-unit and pseudoensemble level, with location being coded by joint populations with identity and outcome, but that identity is coded independently from outcome.

## NAc population activity distinguishes all task phases

Next, we sought to determine how coding of cue features evolved over time. Two main possibilities can be distinguished (*Figure 1B*); a unit coding for a feature such as cue identity could remain persistently active, or a progression of distinct units could activate in sequence. To visualize the distribution of responses throughout our task space and test if this distribution is modulated by cue features, we z-scored the firing rate of each unit, plotted the normalized firing rates of all units aligned to cue-onset, and sorted them according to the time of peak firing rate (*Figure 5*). We did this separately for both the light and sound blocks, and found a nearly uniform distribution of firing fields in task space that was not limited to alignment to the cue (*Figure 5A*). Furthermore, to determine if this population level activity was similar across blocks, we also organized firing during the sound blocks according to the ordering derived from the light blocks. This revealed that while there was some preservation of order, the overall firing was qualitatively different across the two blocks, implying that population activity distinguishes between light and sound blocks.

To control for the possibility that any comparison of trials would produce this effect, we divided each block into two halves and looked at the correlation of the average smoothed firing rates across various combinations of these halves across our cue-onset centered epoch to see if the across-block comparisons were less correlated than the within-block correlations. A linear mixed effects model revealed that within-block correlations (e.g. one half of light trials vs other half of light trials) were higher and more similar than across-block correlations (e.g. half of light trials vs half of sound trials) suggesting that activity in the NAc discriminates across light and sound blocks (mean within-block correlation = 0.38; mean across-block correlation = 0.34; p < .001). This process was repeated for cue location (*Figure 5B*; mean within-block correlation = 0.36; mean across-block correlation = 0.29; p < .001) and cue outcome (*Figure 5C*; mean within-block correlation = 0.35; mean across-block correlation = 0.25; p < .001). Additionally, given that the majority of our units showed an inhibitory response to the cue, we also plotted the firing rates according to the lowest time in firing, and again found some maintenance of order, but largely different ordering across the two blocks (*Figure 5—figure supplement 1*). Together, these results illustrate that NAc coding of task space was not confined to salient events such as cue-onset, but was approximately uniformly distributed throughout the task.

## NAc encoding of cue features persists until outcome

In order to be useful for credit assignment in reinforcement learning, a trace of the cue must be maintained until the outcome, so that information about the outcome can be associated with the outcome-predictive cue (*Figure 1B*). Investigation into the post-approach period during nosepoke revealed units that discriminated various cue features, with some units showing discriminative activity at both cue-onset and nosepoke (*Figure 6*, *Figure 6—figure supplement 1* and *Figure 6—figure supplement 2*). To quantitatively test whether representations of cue features persisted post-approach until the outcome was revealed, we fit sliding window GLMs to the post-approach firing rates of cue-modulated units aligned to both the time of nosepoke into the reward receptacle, and after the outcome was revealed (*Figure 7A,B*, *Figure 7—figure supplement 1A–D* and *Table 1*). This analysis showed that a variety of units discriminated firing according to cue identity (~20% of cue-modulated units) location (~25% of cue-modulated units), and outcome (~25% of cue-modulated units), but not other task parameters, showing that NAc activity discriminates various cue conditions well into a trial.

To determine whether NAc representations of cue features at nosepoke and outcome were encoded by a similar pool of units as during cue-onset, we correlated recoded beta coefficients from the GLMs for a cue feature across time points in the task, and compared the obtained correlations to correlations generated by shuffling unit ordering within a recoded array (*Figures 1B,C* and *7C–F*). This revealed that identity coding was accomplished by a joint population across all three task events (cue-onset and nosepoke: mean r = .05, z-score = 3.47; cue-onset and outcome: mean r = .08, z-score = 5.55; nosepoke and outcome: mean r = .15, z-score = 10.91). Applying this same analysis for cue location revealed a similar pattern for location coding (cue-onset and nosepoke: mean r = .06, z-score = 4.15; cue-onset and outcome: mean r = .09, z-score = 6.40; nosepoke and outcome: mean r = .20, z-score = 14.29). However, outcome coding at cue-onset was separate from coding at nosepoke (mean r = −0.04, z-score = −3.10), and independent from coding at outcome

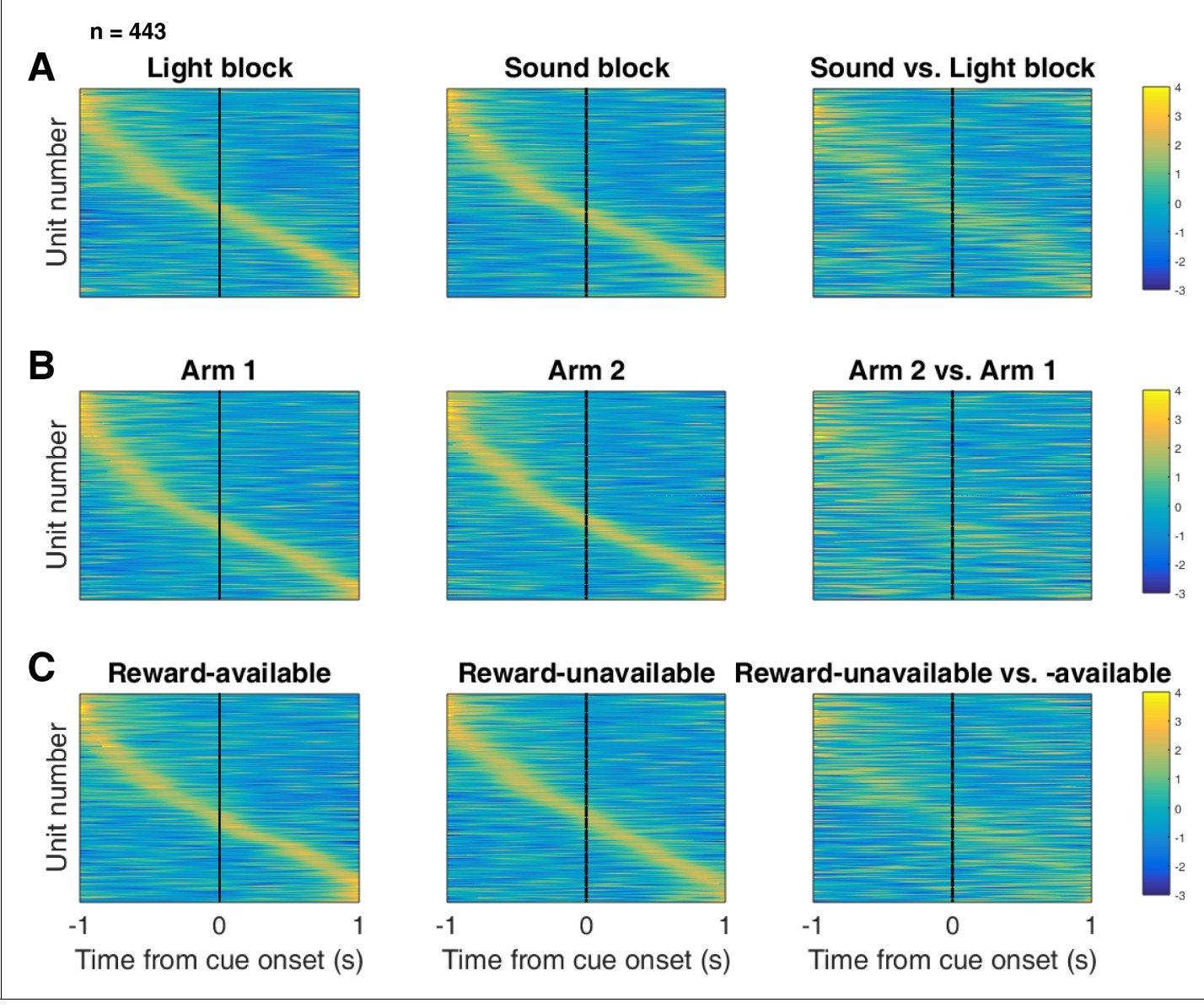

**Figure 5.** Distribution of NAc firing rates across time surrounding cue-onset. Each panel shows normalized (z-score) peak firing rates for all recorded NAc units (each row corresponds to one unit) as a function of time (time 0 indicates cue-onset), averaged across all trials for a specific cue type, indicated by text labels. (A) left: Heat plot showing smoothed normalized firing activity of all recorded NAc units ordered according to the time of their peak firing rate during the light block. Each row is a unit's average activity across time to the light block. Dashed line indicates cue-onset. Notice the yellow band across time, indicating all aspects of visualized task space were captured by the peak firing rates of various units. (A) middle: Same units ordered according to the time of the peak firing rate during the sound block. Note that for both blocks, units tile time approximately uniformly with a clear diagonal of elevated firing rates. (A) right: Unit firing rates taken from the sound block, ordered according to peak firing rate taken from the light block. Note that a weaker but still discernible diagonal persists, indicating partial similarity between firing rates in the two blocks. Colour bar displays relationship between z-score and colour. (B) Same layout as in A, except that the panels now compare two different locations on the track instead of two cue modalities. As for the different cue modalities, NAc units clearly discriminate between locations, but also maintain some similarity across locations, as evident from the visible diagonal in the right panel. Two example locations were used for display purposes; other location pairs showed a similar pattern. (C) Same layout as in A, except that panels now compare reward-available and reward-unavailable trials. Overall, NAc units coded experience on the task, as opposed to being confined to specific task events only. Units from all sessions and animals were pooled for this analysis.

DOI: https://doi.org/10.7554/eLife.37275.011

The following figure supplement is available for figure 5:

**Figure supplement 1.** Distribution of NAc firing rates across time surrounding cue-onset.

DOI: https://doi.org/10.7554/eLife.37275.012

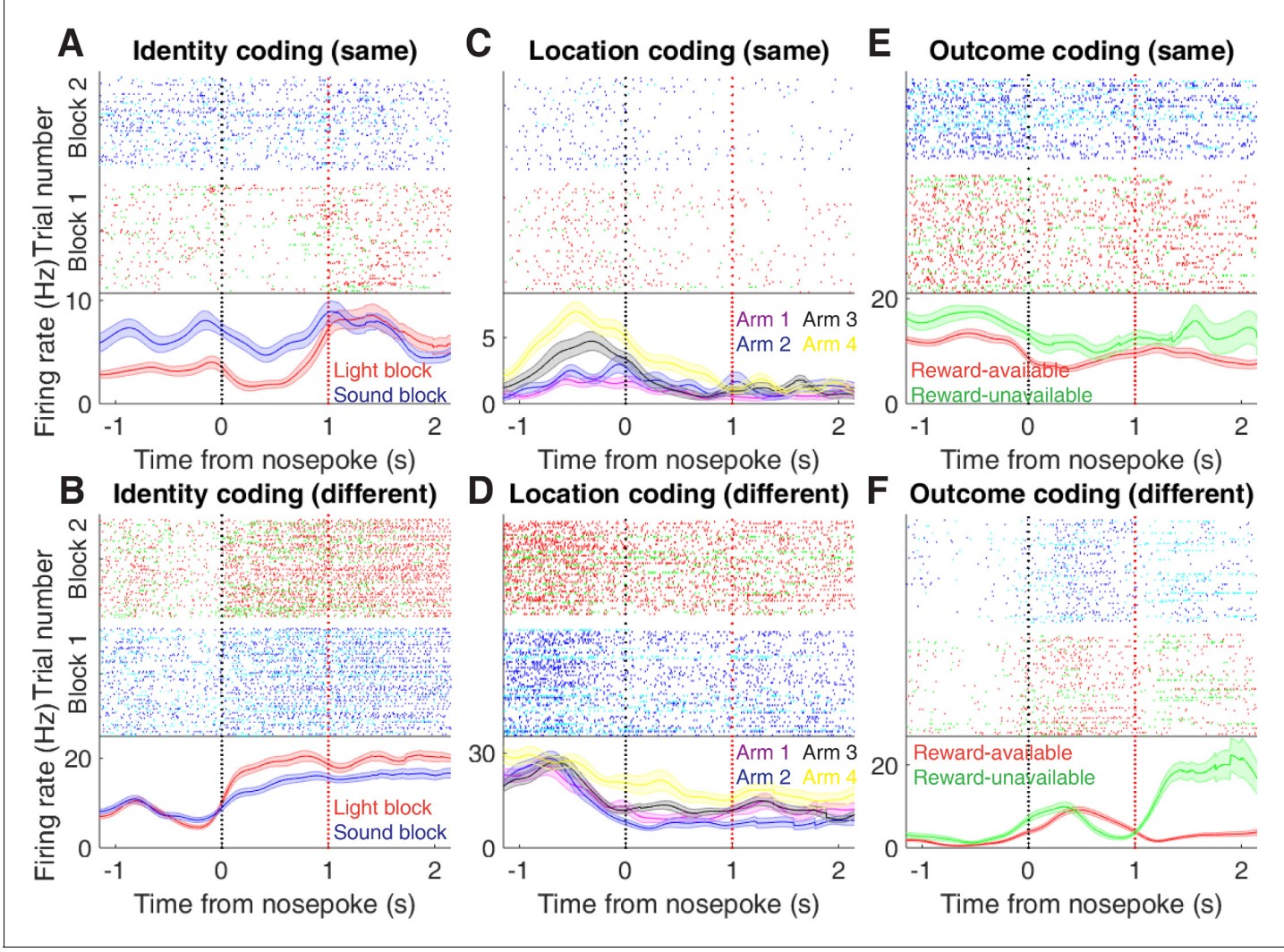

**Figure 6.** Examples of cue-modulated NAc units influenced by cue features at time of nosepoke. (A) Example of a cue-modulated NAc unit that exhibited identity coding at both cue-onset and during subsequent nosepoke hold. Top: raster plot showing the spiking activity across all trials aligned to nosepoke. Spikes across trials are colour coded according to cue type (red: reward-available light; green: reward-unavailable light; navy blue: reward-available sound; light blue: reward-unavailable sound). White space halfway up the raster plot indicates switching from one block to the next. Black dashed line indicates nosepoke. Red dashed line indicates receipt of outcome. Bottom: PETHs showing the average smoothed firing rate for the unit for trials during light (red) and sound (blue) blocks, aligned to nosepoke. Lightly shaded area indicates standard error of the mean. Note this unit showed a sustained increase in firing to sound cues during the trial. (B) An example of a unit that was responsive to cue identity at time of nosepoke but not cue-onset. (C-D) Cue-modulated units that exhibited location coding, at both cue-onset and nosepoke (C), and only nosepoke (D). Each colour in the PETHs represents average firing response for a different cue location. (E-F) Cue-modulated units that exhibited outcome coding, at both cue-onset and nosepoke (E), and only nosepoke (F), with the PETHs comparing reward-available (red) and reward-unavailable (green) trials.
DOI: https://doi.org/10.7554/eLife.37275.013

The following figure supplements are available for figure 6:

**Figure supplement 1.** Expanded examples of cue-modulated NAc units influenced by different cue features at both cue-onset and during subsequent nosepoke hold for *Figure 6A,C,E*, showing firing rate breakdown by: cue type (top PETH), cue identity (top-middle PETH), cue location (bottom-middle PETH), and cue outcome (bottom PETH).
DOI: https://doi.org/10.7554/eLife.37275.014

**Figure supplement 2.** Expanded examples of cue-modulated NAc units influenced by different cue features at time of nosepoke for *Figure 6B,D,F*, showing firing rate breakdown by: cue type (top PETH), cue identity (top-middle PETH), cue location (bottom-middle PETH), and cue outcome (bottom PETH).
DOI: https://doi.org/10.7554/eLife.37275.015

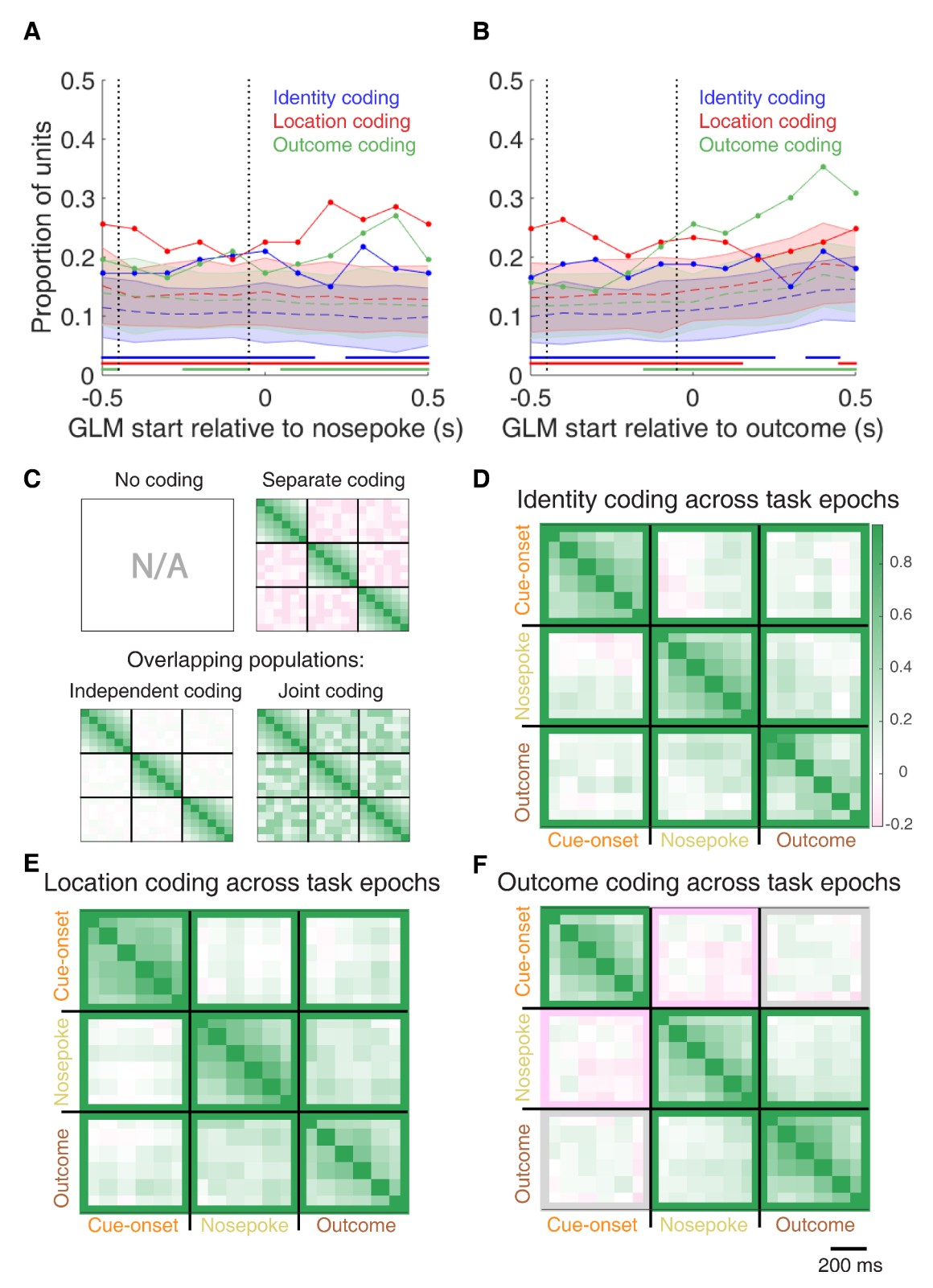

**Figure 7.** Summary of influence of cue features on cue-modulated NAc units at time points surrounding nosepoke and subsequent receipt of outcome. (A-B) Sliding window GLM illustrating the proportion of cue-modulated units influenced by various predictors around time of nosepoke (A), and outcome (B). (A) Sliding window GLM (bin size: 500 ms; step size: 100 ms) demonstrating the proportion of cue-modulated units where cue identity (blue solid line), location (red solid line), and outcome (green solid line) significantly contributed to the model at various time epochs relative to when

*Figure 7 continued on next page*

*Figure 7 continued*

the rat made a nosepoke. Dashed coloured lines indicate the average of shuffling the firing rate order that went into the GLM 100 times. Error bars indicate 1.96 standard deviations from the shuffled mean. Solid lines at the bottom indicate when the proportion of units observed was greater than the shuffled distribution (z-score > 1.96). Points in between the two vertical dashed lines indicate bins where both pre- and post-cue-onset time periods were used in the GLM. (B) Same as A, but for time epochs relative to receipt of outcome after the rat got feedback about his approach. (C-F) Correlation matrices testing the persistence of cue feature coding across points in time. (C) Schematic outlining the possible outcomes for coding of a cue feature across various points in a trial, generated by correlating the recoded beta coefficients from the GLMs and comparing to a shuffled distribution (see text for analysis details). Top left: coding is not present, therefore no comparison is possible. Top right: a cue feature is coded by separate populations of units across time. Displayed is a correlation matrix with each of the nine blocks representing correlations for a cue feature across time bins for two task events from the sliding window GLM, with green representing positive correlations (r > 0), pink negative correlations (r < 0), and white representing significant correlation (r = 0). X- and y-axis have the same axis labels, therefore the diagonal represents the correlation of cue feature against itself at that particular time point (r = 1). Here the large amount of pink in the off-diagonal elements suggests that coding of a cue feature is accomplished by separate populations of units across time. Bottom left: Coding of a cue feature across time occurs in overlapping but independent populations of units, shown here by the abundance of white and relative lack of green and pink in the off-diagonal elements. Bottom right: Coding of a cue feature across time occurs in a joint overlapping population, shown here by the large amount of green in the off-diagonal elements. (D) Correlation matrix showing the correlation of units that exhibited identity coding across time points after cue-onset, nosepoke, and outcome receipt. The window of GLMs used in each block is from the onset of the task phase to the 500 ms window post-onset, in 100 ms steps. Each individual value is for a sliding window GLM within that range, with the scale bar contextualizing step size. Colour bar displays relationship between correlation value and colour. Coloured square borders around each block indicate the result of a comparison of the mean correlation of a block to a shuffled distribution, with pink indicating separate populations (z-score < −1.96), grey indicating overlapping but independent populations, and green indicating joint overlapping populations (z-score > 1.96). (E-F) Same as D, but for location and outcome coding, respectively.

DOI: https://doi.org/10.7554/eLife.37275.016

The following figure supplements are available for figure 7:

**Figure supplement 1.** Summary of influence of cue features on cue-modulated NAc units at time points surrounding nosepoke and subsequent receipt of outcome.

DOI: https://doi.org/10.7554/eLife.37275.017

**Figure supplement 2.** Distribution of NAc firing rates across time surrounding nosepoke for approach trials.

DOI: https://doi.org/10.7554/eLife.37275.018

(mean r = .03, z-score = 1.65), while joint coding was observed between nosepoke and outcome (mean r = .15, z-score = 9.74). Together, these findings show that the NAc maintains representations of cue identity and location by a joint overlapping population throughout a trial, while separate populations of units encode cue outcome before and after a behavioural decision has been made.

To assess overlap among cue features at nosepoke and outcome receipt, we applied the same recoded coefficient analysis (*Figure 7—figure supplement 1E,F*). This revealed joint coding of cue features at the time of nosepoke (cue identity and location: mean r = .12, z-score = 8.26; cue identity and outcome: mean r = .05, z-score = 3.65; cue location and outcome: mean r = .10, z-score = 6.60); while at outcome, identity was coded by a joint population with both location (mean r = .09, z-score = 5.58), and outcome (mean r = .04, z-score = 2.93), and location and outcome were coded by an independent population of units (mean r = .00, z-score = 0.28).

To assess the distributed coding of units for task space around outcome receipt, we aligned normalized peak firing rates to nosepoke onset (*Figure 7—figure supplement 2*). This revealed a clustering of responses around outcome receipt for all cue conditions where the rat would have received reward, in addition to the same pattern of higher within- vs across-block correlations for cue identity (*Figure 7—figure supplement 2A,C*; mean within-block correlation = 0.55; mean across-block correlation = 0.48; p < .001), cue location (*Figure 7—figure supplement 2B,E*; mean within-block correlation = 0.47; mean across-block correlation = 0.41; p < .001), and cue outcome (*Figure 7—figure supplement 2C,F*; mean within-block correlation = 0.51; mean across-block correlation = 0.41; p < .001), further reinforcing that NAc activity distinguishes all task phases.

## Discussion

The main result of the present study is that NAc units encode not only the expected outcome of outcome-predictive cues, but also the identity of such cues (*Figure 1A*). The population of units that coded for cue identity was statistically independent from the population coding for expected outcome at cue-onset (i.e. overlap as expected from chance), while a joint overlapping population

coded for identity and outcome at both nosepoke and outcome receipt (i.e. overlap greater than that expected from chance, *Figure 1C*). Importantly, this identity coding was maintained on approach trials by a similar population of units both during a delay period where the rat held a nosepoke until the outcome was received, and immediately after outcome receipt (*Figure 1B,C*). Cue identity information was also present at the population level, as indicated by high classification performance based on pseudoensembles. More generally, NAc unit activity profiles were not limited to salient task events such as the cue, nosepoke and outcome, but were distributed more uniformly throughout the task. This temporally distributed activity differed systematically between cue identities, expected outcomes and locations. We discuss these observations and their implications below.

## Identity coding

Our finding that NAc units can discriminate between different outcome-predictive stimuli with similar motivational significance (i.e. encode cue identity) expands upon an extensive rodent literature examining NAc correlates of conditioned stimuli (*Ambroggi et al., 2008*; *Atallah et al., 2014*; *Bissonette et al., 2013*; *Cooch et al., 2015*; *Day et al., 2006*; *Dejean et al., 2017*; *Goldstein et al., 2012*; *Ishikawa et al., 2008*; *Lansink et al., 2012*; *McGinty et al., 2013*; *Nicola et al., 2004*; *Roesch et al., 2009*; *Roitman et al., 2005*; *Saddoris et al., 2011*; *Setlow et al., 2003*; *Sugam et al., 2014*; *West and Carelli, 2016*; *Yun et al., 2004*). Perhaps the most comparable work in rodents comes from a study that found a subset of NAc units that modulated their firing for an odor when it predicted distinct but equally valued rewards (*Cooch et al., 2015*). The present study is complementary to such *outcome identity* coding, in showing that NAc units encode *cue identity* in addition to the reward it predicts (*Figure 1A*). *Setlow et al. (2003)* paired two distinct odor cues with appetitive and aversive odor cues respectively in a Go/NoGo task, such that cue identity and cue outcome were linked. Although reversal sessions were run that uncoupled identity and outcome, the resulting changes in reinforcement history and behavioural performance precluded a clear test of cue identity coding. Thus, our study is distinct in asking how different cues encoding the same anticipated outcome are encoded. Our results suggest that the NAc dissociates cue identity representations at multiple levels of analysis (e.g. single-unit and population) even when the motivational significance of these stimuli is identical. Viewed within the neuroeconomic framework of decision making, functional magnetic resonance imaging (fMRI) studies have found support for NAc representations of *offer value*, a domain-general common currency signal that enables comparison of different attributes such as reward identity, effort, and temporal proximity (*Bartra et al., 2013*; *Levy and Glimcher, 2012*; *Peters and Büchel, 2009*; *Sescousse et al., 2015*). Our study adds to a growing body of electrophysiological research that suggests the view of the NAc as a value centre, while informative and capturing major aspects of NAc processing, neglects additional contributions of NAc to learning and decision making such as the offer (cue) identity signal reported here.

Our analyses were designed to eliminate several potential alternative interpretations to cue identity coding. Because the different cues were separated into different blocks, units that discriminated between cue identities could instead be encoding time or other slowly-changing quantities. We excluded this possible confound by excluding units that showed a drift in firing between the first and second half within a block. Additionally, we included time as a nuisance variable in our GLMs, to exclude firing rate variance in the remaining units that could be attributed to this confound. Furthermore, we found that the temporally evolving firing rate throughout a trial was more strongly correlated within a block than across blocks. However, the possibility remains that instead of, or in addition to, stimulus identity, these units encode a preferred context, or even a macro-scale representation of progress through the session. Indeed, encoding of the current strategy could be an explanation for the presence of pre-cue identity coding (*Figure 4A*), as well as for the differential distributed coding of task structure across blocks observed in the current study (*Figure 5*).

An overall limitation of the current study is that rats were never presented with both sets of cues simultaneously, and were not required to switch strategies between multiple sets of cues (this was attempted in behavioural pilots, but animals took several days of training to successfully switch strategies). Additionally, our recordings were done during performance on the well-learned behaviour, and not during the initial acquisition of the cue-outcome relationships when an eligibility trace would be most useful. Thus, it is unknown to what extent the cue identity encoding we observed is behaviourally relevant, although extrapolating data from other work (*Sleezer et al., 2016*) suggests that

cue identity coding would be modulated by relevance. Furthermore, NAc core lesions have been shown to impair shifting between different behavioural strategies (*Floresco et al., 2006*), and it is possible that selectively silencing the units that prefer responding for a given modality or rule would impair performance when the animal is required to use that information, or artificial enhancement of those units would cause them to use the rule when it is the inappropriate strategy.

## NAc activity provides a rich task representation beyond reward alone

Beyond coding of cue identity, we found several other notable features of NAc activity. First, a substantial number of cue-modulated units was differentially active depending on location, consistent with previous reports (*Lavoie and Mizumori, 1994*; *Mulder et al., 2005*; *Strait et al., 2016*; *Wiener et al., 2003*). However, it is notable that in our task, location is explicitly uninformative about reward, yet coding of this uninformative variable persists. This is unlike previous work of location coding in the dorsolateral striatum, which was present when location was predictive of reward, and absent when it was uninformative (*Schmitzer-Torbert and Redish, 2008*). Persistent coding of location in the NAc is likely attributable to inputs from the hippocampus (*Lansink et al., 2016*; *Sjulson et al., 2018*; *Tabuchi et al., 2000*; *van der Meer and Redish, 2011*); speculatively, this coding may map onto a bias in credit assignment, such that motivationally relevant events are likely to be associated with the locations where they occur.

A second striking feature of NAc activity evident from this task is that NAc units were not only active at salient events such as cue presentation, nosepoking, and feedback about the outcome, but distributed their activity throughout a trial (*Figure 5*). This observation is consistent with previous work reporting that NAc units can signal progress through a sequence of cues and/or actions (*Atallah et al., 2014*; *Berke et al., 2009*; *Khamassi et al., 2008*; *Lansink et al., 2012*; *Mulder et al., 2004*; *Shidara et al., 1998*) and reminiscent of similar observations in the ventral pallidum (*Tingley et al., 2014*) to which the NAc projects. Extending this previous work, we show that the specific pattern of NAc units throughout a trial can be modified by task variables such as cue identity. This richer view of NAc activity recalls a dynamical systems perspective, in which different task conditions correspond to different trajectories in a neural state space (e.g. *Buonomano and Maass, 2009*; *Shenoy et al., 2013*). In any case, this view of NAc activity provides a substantially richer picture than that expected from encoding of reward-related variables alone.

## Functional relevance of cue identity coding

One possible function of cue identity coding is to support contextual modulation of the motivational relevance of specific cues. A context can be understood as a particular mapping between specific cues and their outcomes: for instance, in context one cue A but not cue B is rewarded, whereas in context two cue B but not cue A is rewarded. Successfully implementing such contextual mappings requires representation of the cue identities. Indeed, *Sleezer et al. (2016)* recorded NAc responses during the Wisconsin Card Sorting Task, a common set-shifting task used in both the laboratory and clinic, and found units that preferred firing to stimuli when a certain rule, or rule category was currently active. Further support for a modulation of NAc responses by strategy comes from an fMRI study that examined blood-oxygen-level dependent (BOLD) levels during a set-shifting task (*FitzGerald et al., 2014*). In this task, participants learned two sets of stimulus-outcome contingencies, a visual set and an auditory set. During testing they were presented with both simultaneously, and the stimulus dimension that was relevant was periodically shifted between the two. It was found that bilateral NAc activity reflected value representations for the currently relevant stimulus dimension, and not the irrelevant stimulus dimension. Given that BOLD activity is thought to reflect the processing of incoming and local information, and not spiking output (*Logothetis et al., 2001*), it is possible that the relevance-gated value representations observed by *FitzGerald et al. (2014)* are integrated with the relevant identity coding in the output of the NAc, as observed in the current study.

A different potential role for cue identity coding is in learning to associate rewards with reward-predictive features of the environment, a process referred to as *credit assignment* in the reinforcement learning literature (*Sutton and Barto, 1998*). Maladaptive decision making, as occurs in schizophrenia, addiction, Parkinson's disease and others can result from dysfunctional reward prediction errors (RPEs) and value signals (*Frank et al., 2004*; *Gradin et al., 2011*; *Maia and Frank, 2011*).

This view has been successful in explaining both positive and negative symptoms in schizophrenia, and deficits in learning from feedback in Parkinson's (*Frank et al., 2004*; *Gradin et al., 2011*). However, the effects of RPE and value updating are contingent upon encoding of preceding action and cue features, the eligibility trace (*Lee et al., 2012*; *Sutton and Barto, 1998*). Value updates can only be performed on these aspects of preceding experience that are encoded when the update occurs. Therefore, maladaptive learning and decision making can result from not only aberrant RPEs but also from altered cue feature encoding. For instance, on this task the environmental stimulus that signalled the availability of reward was conveyed by two distinct cues that were presented in four locations. Although in our current study, the location and identity of the cue did not require any adjustments in the animal's behaviour, we found coding of these features alongside the expected outcome of the cue that could be the outcome of credit assignment computations computed upstream (*Akaishi et al., 2016*; *Asaad et al., 2017*; *Chau et al., 2015*; *Noonan et al., 2017*). Identifying neural coding related to an aspect of credit assignment is important as inappropriate credit assignment could be a contributor to conditioned fear overgeneralization seen in disorders with pathological anxiety such as generalized anxiety disorder, post-traumatic stress disorder, and obsessive-compulsive disorder (*Kaczkurkin et al., 2017*; *Kaczkurkin and Lissek, 2013*; *Lissek et al., 2014*), and delusions observed in disorders such as schizophrenia, Alzheimer's and Parkinson's (*Corlett et al., 2010*; *Kapur, 2003*). Thus, our results provide a starting point for studies of the neural basis of credit assignment, and the extent and specific manner in which this process fails in syndromes such as schizophrenia, obsessive-compulsive disorder, and others.

## Materials and methods

### Subjects

A sample size of 4 adult male Long-Evans rats (Charles River, Saint Constant, QC) from an a priori determined sample of 5 were used as subjects (one rat was excluded from the data set due to poor cell yield). Rats were individually housed with a 12/12 hr light-dark cycle, and tested during the light cycle. Rats were food restricted to 85–90% of their free feeding weight (weight at time of implantation was 440–470 g), and water restricted 4–6 hr before testing. All experimental procedures were approved by the University of Waterloo Animal Care Committee (protocol# 11–06) and carried out in accordance with Canadian Council for Animal Care (CCAC) guidelines.

### Overall timeline

Each rat was first handled for seven days during which they were exposed to the experiment room, the sucrose solution used as a reinforcer, and the click of the sucrose dispenser valves. Rats were then trained on the behavioural task (described in the next section) until they reached performance criterion. At this point they underwent hyperdrive implantation targeted at the NAc. Rats were allowed to recover for a minimum of five days before being retrained on the task, and recording began once performance returned to pre-surgery levels. Upon completion of recording, animals were gliosed, euthanized and recording sites were histologically confirmed.

### Behavioural task and training

The behavioural apparatus was an elevated, square-shaped track (100 × 100 cm, track width 10 cm) containing four possible reward locations at the end of track 'arms' (*Figure 2A*). Rats initiated a *trial* by triggering a photobeam located 24 cm from the start of each arm. Upon trial initiation, one of two possible light cues (L1, L2), or one of two possible sound cues (S1, S2), was presented that signalled the presence (*reward-available trial*, L1+, S1+) or absence (*reward-unavailable trial*, L2-, S2-) of a 12% sucrose water reward (0.1 mL) at the upcoming reward site. A trial was classified as an *approach trial* if the rat turned left at the decision point and made a nosepoke at the reward receptacle (40 cm from the decision point), while a trial was classified as a *skip trial* if the rat instead turned right at the decision point and triggered the photobeam to initiate the next trial. A trial was labelled *correct* if the rat approached (i.e. nosepoked) on reward-available trials, and skipped (i.e. did not nosepoke) on reward-unavailable trials. On reward-available trials there was a 1 s delay between a nosepoke and subsequent reward delivery. *Trial length* was determined by measuring the length of time from cue-onset until nosepoke (for approach trials), or from cue-onset until the

start of the following trial (for skip trials). Trials could only be initiated through clockwise progression through the series of arms, and each entry into the subsequent arm on the track counted as a trial. Cues were present until 1 s after outcome receipt on approach trials, and until initiating the following trial on skip trials.

Each session consisted of both a *light block* and a *sound block* with 100 trials each. Within a block, one cue signalled reward was available on that trial (L1+ or S1+), while the other signalled reward was not available (L2- or S2-). Light block cues were a flashing white light, and a steady yellow light. Sound block cues were a 2 kHz sine wave (low) and a 8 kHz sine wave (high) whose amplitude was modulated from 0 to maximum by a 2 Hz sine wave. Outcome-cue associations were counterbalanced across rats, for example for some rats L1+ was the flashing white light, and for others L1+ was the steady yellow light. The order of cue presentation was pseudorandomized so that the same cue could not be presented more than twice in a row. Block order within each day was also pseudorandomized, such that the rat could not begin a session with the same block for more than two days in a row. Each session consisted of a 5 min pre-session period on a pedestal (a terracotta planter filled with towels), followed by the first block, then the second block, then a 5 min post-session period on the pedestal. For approximately the first week of training, rats were restricted to running in the clockwise direction by presenting a physical barrier to running counterclockwise. Cues signalling the availability and unavailability of reward, as described above, were present from the start of training. Rats were trained for 200 trials per day (100 trials per block) until they discriminated between the reward-available and reward-unavailable cues for both light and sound blocks for three consecutive days, according to a chi-square test rejecting the null hypothesis of equal approaches for reward-available and reward-unavailable trials, at which point they underwent electrode implant surgery.

## Surgery

Surgical procedures were as described previously (*Malhotra et al., 2015*). Briefly, animals were administered analgesics and antibiotics, anesthetized with isoflurane, induced with 5% in medical grade oxygen and maintained at 2% throughout the surgery (~ 0.8 L/min). Rats were then chronically implanted with a 'hyperdrive' consisting of 20 independently drivable tetrodes, with four designated as referencetetrodes, and the remaining 16 either all targeted to the right NAc (AP+ 1.4 mm and ML+ 1.6 mm relative to bregma; *Paxinos and Watson, 1998*), or 12 in the right NAc and four targeted to the mPFC (AP +3.0 mm and ML +0.6 mm, relative to bregma; only data from NAc tetrodes was analysed). Following surgery, all animals were given at least five days to recover while receiving post-operative care, and tetrodes were lowered to the target (DV −6.0 mm) before being reintroduced to the behavioural task.

## Data acquisition and preprocessing

After recovery, rats were placed back on the task for recording. NAc signals were acquired at 20 kHz with a RHA2132 v0810 preamplifier (Intan) and a KJE-1001/KJD-1000 data acquisition system (Amplipex). Signals were referenced against a tetrode placed in the corpus callosum above the NAc. Candidate spikes for sorting into putative single units were obtained by band-pass filtering the data between 600–9000 Hz, thresholding and aligning the peaks UltraMegaSort2k, (*Hill et al., 2011*). Spike waveforms were then clustered with KlustaKwik using energy and the first derivative of energy as features, and manually sorted into units (MClust 3.5, A.D. Redish et al., http://redishlab.neuroscience.umn.edu/MClust/MClust.html). Isolated units containing a minimum of 200 spikes within a session were included for subsequent analysis. Units were classified as FSIs by an absence of interspike intervals (ISIs) > 2 s, while MSNs had a combination of ISIs > 2 s and phasic activity with shorter ISIs (*Barnes et al., 2005*; *Atallah et al., 2014*).

## Data analysis
### Behaviour

To determine if rats distinguished behaviourally between the reward-available and reward-unavailable cues (*cue outcome*), we generated linear mixed effects models to investigate the relationships between cue type and the proportion of trials approached, with *cue outcome* (reward available or not) and *cue identity* (light or sound) as fixed effects, and the addition of an intercept for rat identity

as a random effect. For each cue, the average proportion of trials approached for a session was used as the response variable. Contribution of cue outcome to behaviour was determined by comparing the full model to a model with cue outcome removed. To assess within-session learning we divided each block into two halves, and compared a model including a block half variable to a null model excluding this variable, to see if adding block half improved prediction of overall behavioural performance.

## Neural data

Given that some of our analyses compare firing rates across time, particularly comparisons across blocks, we sought to exclude units with unstable firing rates that would generate spurious results reflecting a drift in firing rate over time unrelated to our task. We used a multipronged strategy to address this potential confound. As a first step, we ran a Mann-Whitney U test comparing the cue-modulated firing rates for the first and second half of trials within a block, and excluded 99 of 443 units from analysis that showed a significant change for either block, leaving 344 units for further analyses by our GLM. Furthermore, we included time (trial number) as a nuisance variable in our GLMs to control for firing rate variance accounted for by this confound (see below). To investigate the contribution of different cue features (*cue identity*, *cue location* and *cue outcome*) on the firing rates of NAc single units, we first determined whether firing rates for a unit were modulated by the onset of a cue by collapsing across all cues and comparing the firing rates for the 1 s preceding cue-onset with the 1 s following cue-onset. Single units were considered to be *cue-modulated* if a Wilcoxon signed-rank test comparing pre- and post-cue firing was significant at $p < .01$. Cue-modulated units were then classified as either increasing or decreasing if the post-cue activity was higher or lower than the pre-cue activity, respectively.

To determine the relative contribution of different task parameters to firing rate variance (as in *Figure 4A*, *Figure 4—figure supplement 1*), a forward selection stepwise GLM using a Poisson distribution for the response variable was fit to each cue-modulated unit, using data from every trial in a session. Cue identity (light block, sound block), cue location (arm 1, arm 2, arm 3, arm 4), cue outcome (reward-available, reward-unavailable), behaviour (approach, skip), trial length, trial number, and trial history (reward availability on the previous two trials) were used as predictors, with firing rate as the response variable. The GLMs were fit using a 500 ms sliding window moving in 100 ms steps centered at 250 ms pre-cue (so no post-cue activity was included) to centered at 750 ms post-cue, such that 11 different GLMs were fit for each unit, tracking the temporal dynamics of the influence of task parameters on firing rate around the onset of the cue. Units were classified as being modulated by a given task parameter if addition of the parameter significantly improved model fit using deviance as the criterion ($p < .01$), and the total proportion of cue-modulated units influenced by a task parameter was counted for each time bin. A comparison of the R-squared value between the final model and the final model minus the predictor of interest was used to determine the amount of firing rate variance explained by the addition of that predictor for a given unit. To control for the amount of units that would be affected by a predictor by chance, we shuffled the trial order of firing rates for a particular unit within a time bin, ran the GLM with the shuffled firing rates, counted the proportion of units encoding a predictor, and took the average of this value over 100 shuffles. We then calculated how many standard deviations the observed proportion was from the mean of the shuffled distribution. For this and all subsequent shuffle analyses, we used a z-score of greater than 1.96 or less than −1.96 as a marker of significance. To further control for whether outcome coding could be attributed to subsequent behavioural variability at the choice point, we ran our cue-onset GLM for approach trials only.

To get a sense of the predictive power of these cue feature representations we trained a classifier using firing rates from a pseudoensemble comprised of our 133 cue-modulated units (*Figure 4B*). We created a matrix of firing rates for each time epoch surrounding cue-onset where each row was an observation representing the firing rate for a trial, and each column was a variable representing the firing rate for a given unit. Trial labels (classes) were each condition for a cue feature (e.g. light and sound for cue identity), making sure to align trial labels across units. We then ran LDA on these matrices, using 10-fold cross validation to train the classifier on 90% of the trials and testing its predictions on the held out 10% of trials, and repeated this approach to get the classification accuracy for 100 iterations. To test if the classification accuracy was greater than chance, we shuffled the

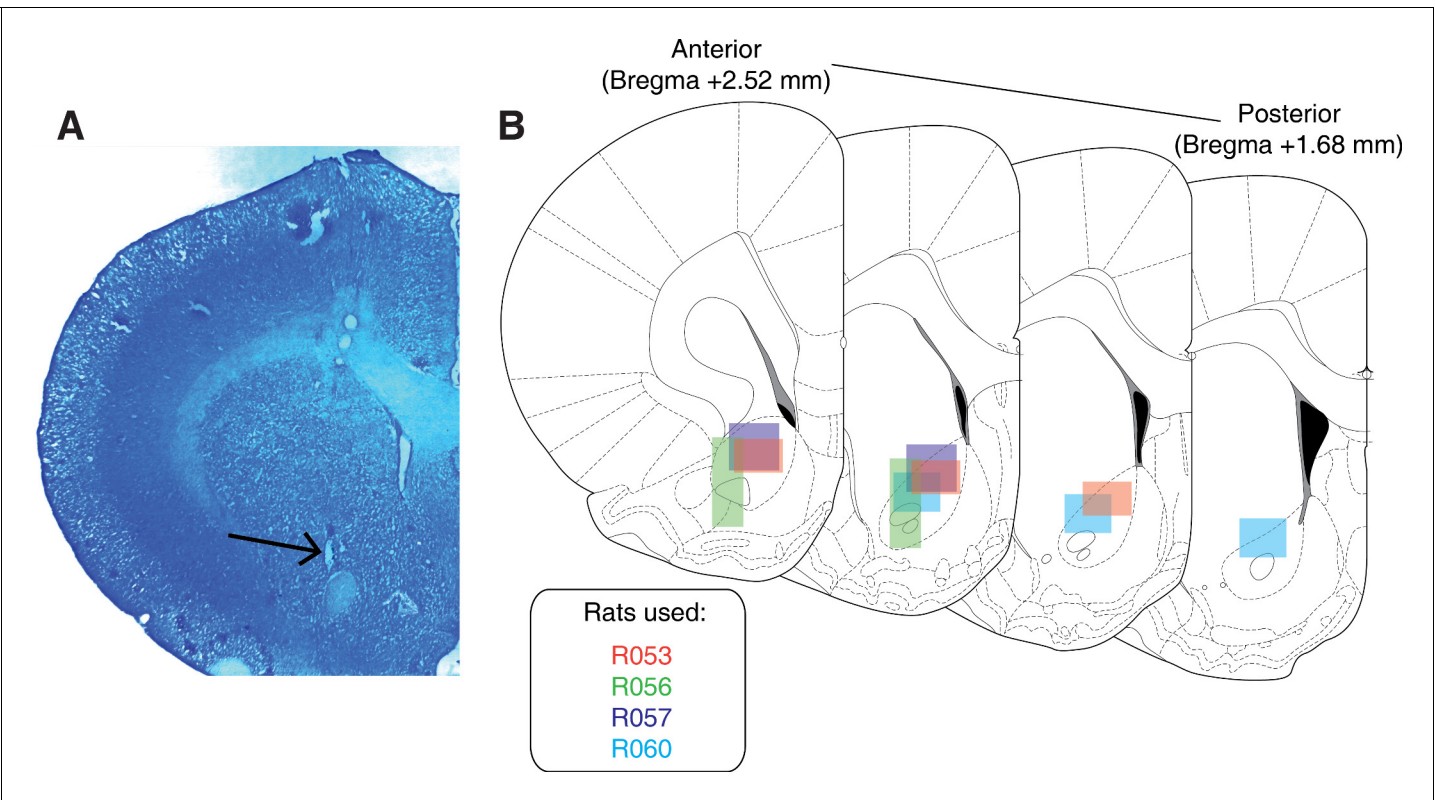

**Figure 8.** Histological verification of recording sites. Upon completion of experiments, brains were sectioned and tetrode placement was confirmed. (**A**) Example section from R060 showing a recording site in the NAc core just dorsal to the anterior commissure (arrow). (**B**) Schematic showing recording areas for all subjects.

DOI: https://doi.org/10.7554/eLife.37275.019

order of firing rates for each unit before we trained the classifier. We repeated this for 100 shuffled matrices for each time point, and calculated how many z-scores the mean classification rate of the observed data was from the mean of the shuffled distribution.

To determine the degree to which coding of cue identity, cue location, and cue outcome overlapped within units we correlated the recoded beta coefficients from the GLMs for the cue features (*Figure 4C,D*). Specifically, we generated an array for each cue feature at each point in time, where for all cue-modulated units we coded a '1' if the cue feature was a significant predictor in the final model, and '0' if it was not. We then correlated an array of the coded 0 s and 1 s for one cue feature with a similar array for another cue feature, repeating this process for all post cue-onset sliding window combinations. The NAc was determined as coding a pair of cue features in a) separate populations of units if there was a significant negative correlation ($r < 0$), b) an independently coded overlapping population of units if there was no significant correlation ($r = 0$), or c) a jointly coded overlapping population of units if there was a significant positive correlation ($r > 0$). To summarize the correlation matrices generated from this analysis, we shuffled the unit ordering for each array 100 times, took the mean of the 36 correlations for a block comparison for each of the 100 shuffles for an analysis window, and used the mean and standard deviation of these shuffled correlation averages to compare to the mean of the comparison block for the actual data.

To better visualize responses to cues and enable subsequent population level analyses (as in *Figures 3* and *5*), spike trains were convolved with a Gaussian kernel ($\sigma = 100$ ms), and peri-event time histograms (PETHs) were generated by taking the average of the convolved spike trains across all trials for a given task condition. To visualize NAc representations of task space within cue conditions, normalized spike trains for all units were ordered by the location of their maximum or minimum firing rate for a specified cue condition (*Figure 5*). To compare representations of task space across cue conditions for a cue feature, the ordering of units derived for one condition (e.g. light block) was

then applied to the normalized spike trains for the other condition (e.g. sound block). To assess whether the task distributions were different across cue conditions, we split each cue condition into two halves, controlling for the effects of time by shuffling trial ordering before the split, and calculated the correlation of the temporally evolving smoothed firing rate across each of these halves, giving us six correlation values for each unit. We then concatenated these six values across all 443 units to give us an array of 2658 correlation coefficients. We then fit a linear mixed effects model, trying to predict these block comparison correlations with comparison type (e.g. 1st half of light block vs. 1st half of sound block) as a fixed-effect term, and unit number as a random-effect term. Comparison type is nominal, so dummy variables were created for the various levels of comparison type, and coefficients were generated for each condition, referenced against one of the within-block comparison types (e.g. 1st half of light block vs. 2nd half of light block). The NAc was considered to discriminate across cue conditions if across-block correlations were lower than within-block correlations. Additionally, we ran a model comparison between the above model and a null model with just unit number, to see if adding comparison type improved model fit.

To identify the responsivity of units to different cue features at the time of nosepoke into a reward receptacle, and subsequent reward delivery, the same cue-modulated units from the cue-onset analyses were analysed at the time of nosepoke and outcome receipt using identical analysis techniques for all approach trials (*Figures 6* and *7*). To compare whether coding of a given cue feature was accomplished by the same or distinct population of units across time epochs, we ran the recoded coefficient correlation that was used to assess the degree of overlap among cue features within a time epoch. All analyses were completed in MATLAB R2015a, and the code and pre-processed data files are available on our public GitHub repository (*Gmaz, 2018*; copy archived at https://github.com/elifesciences-publications/vStrCueCodingPaper).

## Histology

Upon completion of the experiment, recording channels were gliosed by passing 10 µA current for 10 s and waiting 5 days before euthanasia, except for rat R057 whose implant detached prematurely. Rats were anesthetized with 5% isoflurane, then asphyxiated with carbon dioxide. Transcardial perfusions were performed, and brains were fixed and removed. Brains were sectioned in 50 µm coronal sections and stained with thionin. Sections were visualized under light microscopy, tetrode placement was determined, and electrodes with recording locations in the NAc were analysed (*Figure 8*).

---

# Additional information

## Funding

| Funder | Author |
| --- | --- |
| Natural Sciences and Engineering Research Council of Canada | Jimmie M Gmaz<br>Matthijs AA van der Meer |

The funders had no role in study design, data collection and interpretation, or the decision to submit the work for publication.

## Author contributions

Jimmie M Gmaz, Data curation, Software, Formal analysis, Funding acquisition, Investigation, Visualization, Methodology, Writing—original draft, Writing—review and editing; James E Carmichael, Investigation, Methodology, Discussion; Matthijs AA van der Meer, Conceptualization, Software, Supervision, Funding acquisition, Methodology, Writing—original draft, Writing—review and editing

## Author ORCIDs

Jimmie M Gmaz ⓘD http://orcid.org/0000-0001-6883-5811
Matthijs AA van der Meer ⓘD http://orcid.org/0000-0002-2206-4473

## Ethics

Animal experimentation: All experimental procedures were approved by the the University of Waterloo Animal Care Committee (protocol# 11-06) and carried out in accordance with Canadian Council for Animal Care (CCAC) guidelines.

## Decision letter and Author response

Decision letter https://doi.org/10.7554/eLife.37275.024
Author response https://doi.org/10.7554/eLife.37275.025

# Additional files

## Supplementary files

• Transparent reporting form
DOI: https://doi.org/10.7554/eLife.37275.020

## Data availability

Preprocessed data and data analysis code, sufficient to reproduce the results in the paper, are available on this public GitHub repository: https://github.com/jgmaz/vStrCueCodingPaper (commit 56c5f52); copy archived at https://github.com/elifesciences-publications/vStrCueCodingPaper.

The following dataset was generated:

| Author(s) | Year | Dataset title | Dataset URL | Database, license, and accessibility information |
|---|---|---|---|---|
| Jimmie M Gmaz | 2018 | vStrCueCodingPaper | https://github.com/jgmaz/vStrCueCodingPaper | Publicly available at Github (https://github.com) |

# Acknowledgements

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
