## [Decision Letter]

Thank you for submitting your article "Persistent coding of outcome-predictive cue features in the rat nucleus accumbens" for consideration by *eLife*. Your article has been reviewed by three peer reviewers, including Geoffrey Schoenbaum as the Reviewing Editor, and the evaluation has been overseen by Michael Frank as the Senior Editor. The reviewers have opted to remain anonymous.

The reviewers have discussed the reviews with one another and the Reviewing Editor has drafted this decision to help you prepare a revised submission.

Summary:

The article describes single unit activity in rat NAc during retrieval on a maze in which reward was available on 4 arms off a square of alleys, signaled by visual or auditory cues in different trial blocks. The main finding is that, contrary to popular ideas, the NAc neurons exhibited as much differential activity to the cues and locations as to the presence or absence of reward. Further the correlates related to cues were orthogonal to information about reward and also persisted into the reward period, providing a potential eligibility trace or substrate for credit assignment underlying learning. The paper adds to a growing number of studies showing that information in this area is much richer than simple theories suggest.

Essential revisions:

The reviews were overall positive, but two reviewers had significant trouble parsing the rich data presentation to see the key analysis. Thus the truly essential revision is to highlight the critical comparisons that show globally how the encoding of features is dissociable from the encoding of reward. Is this information truly separate in the population, merely orthogonal? Embedded in the reward coding? This can be done with the current single unit analysis, but an interesting adjunct might be to analyze the neurons as populations and remove variance related to the different variables. This or similar analyses would be a nice addition to allow a clear demonstration of what information is in the activity if the authors think it is possible. Alternatively simply clarifying the current approach, maybe at the reduction of some of the anecdotal description might do it.

In addition to this, there are some potential confounds raised that the authors should address. The problem with response direction is important and also drift with or across block and how this is handled should be made clearer.

Lastly the reviewers thought that the authors might consider more the implications of their data for accounts of this area as a key region for calculating economic value.

Reviewer #1:

This paper reports on neural correlates in rat NAc during performance of a square maze task in which auditory or visual cues delivered in different blocks of trials signaled reward or non-reward down alleys at each corner of the square. The authors report that, by GLM analysis, significant numbers of NAc neurons were engaged by the cues, the locations in the arms, and the likelihood of reward. Moreover the representation of non-reward information was judged to be independent of or orthogonal to representation of information about reward. This result is generally contrary to the view that the NAc is a value-coding center, either in the tradition of Mogenson or in the more recent framework of computational or neuroeconomic models. This joins a growing number of studies suggesting associative coding in this area is more complex than generally assumed. Overall I liked the study. The task is interesting, and the analyses are diverse and fairly exhaustive, considering the data from many angles. Actually my main criticism is probably that this makes the main point difficult to maintain and keep track of. I felt like there was too much description of the various anecdotal results in the text and figures and not enough clear demonstration of the independence or orthogonal nature of the cue representations from value. I have a couple of suggestions for this, but I'm sure the authors will also have ideas.

For me the main point to make clear is that the neural activity contains information about the cues, contexts, location etc. that is not driven by reward. Note this could be independent, orthogonal, or even integrated. Currently the authors show this now with their GLM analysis of individual unit firing and by comparing proportions of neurons doing different things and amount of overlap by Chi-Squared. I think this is fine, however I think it leads to some complexity and confusion. For example, they say "NAc encodes features of reward-predicting cues separate from expected outcome" referencing Figure 4A among others. While this neuron might show this, it might also (I think) be reflecting the conjunction of sound and reward. Which do they mean? Then just after this they say, "overlap of coding of cue features within units was not different than expected by chance according to Chi-Squared tests, suggesting for integrated coding across various aspects of a cue". It seems to me that this shows a lack of integrated coding if it is no more than chance? I was left a bit unsure about how independent representations were.

I wonder if the authors can be much more explicit with the Venn diagrams and comparisons about what is above chance for the unit analysis. Then I think it would be useful to apply an analysis across units – using pseudoensembles – to ask what information is present with and without value present. This would allow the authors to demonstrate clearly that the units, either individually or as a population, contain information about features that is orthogonal to the reward. A similar strategy could be used to show whether the feature information is also integrated or not I think.

Reviewer #2:

This paper addresses the simple question of whether the firing of single neurons in nucleus accumbens is selective for the identity of a stimulus during cue presentation and the time between the decision and outcome. The answer is that they appear to encode location in the maze, sensory properties of stimuli and the prediction of reward or no reward. The task is a maze where rats circle around for food. There are four decision points where they have to decide to take a left for possible food or take a right and go on to the next decision point. At each decision-point a noise or light signals if reward is available for the left turn. Auditory and visual stimuli are used in different blocks. I have several issues with the paper.

- I think the task is overly complicated to answer this question. There are too many confounds and it is not clear when animals hear/see stimuli. Auditory and visual stimuli are presented in different blocks, which leads to confounds related to context, current strategy (authors admit this one) and drift. There should have been a third block during which the modality of the cue was the same as the first block to show genuine cue selectivity. Over, it was not clear to me why auditory and visual stimuli were not used in the same block of trials. Such a task design would have been far more convincing.

- Another confound that resulted from task design is that rewards are only provided for left turns. Right turns only led to the next decision point. Thus neurons that appear to be encoding left and right might be signaling no reward or the possibility of reward.

- I did not find the plotted histograms overly informative. Many of them simply plot firing rates associated with the trial-types that the neurons were selected for. It would be more useful to show us those trials that the neurons were selective for but also break trials down in the other parameters. For example, for neurons that prefer light blocks over sound blocks (or vice versa), we should be shown those trial-types but also trials broken down by the decision the animal made and if reward was available. This strategy should be done for cells that are selective for block, reward, and arm, at the single neuron and population level.

- I'm not convinced that what we are seeing in single neuron and population histograms is not due to drift. Too much of the selectivity is observed before cue onset. This is a problem for all cell types but even a bigger problem for neurons selective for reward availability.

- Along this same issue there should be figures that show how selectivity emerged in both trial blocks.

- Overall figures are too small making it very difficult to see things. For example the differently colored rasters are impossible to see. They should be bigger. Also, the information carried in those rasters related to trial-types should be shown in the average line plots below as suggested above.

- Percentage should be provided in the table. Also, some indication if counts are significantly more than chance should be added to the table.

- The point of the results illustrated in Figure 7 and 11 is not clear. Scatter plots (where each dot is a neuron), that plot firing on light blocks versus sound blocks would better show if neurons are or are not encoding expected reward across modalities.

- It is not always clear what trial types are being analyzed throughout the paper; correct trials only?

- Liner models are used but is firing normal?

- These neurons might be involved in credit assignment but I think the authors' claims are too far reaching. There is no inactivation. There is no link between firing and behavior, and rats are already well trained on the task.

- How long were cues on?

- Early work by Schultz examining these issues should be mentioned.

- These effects are not that novel, as the authors point out, because others have already shown odor selectivity. This is true even in reversal tasks where the different cues predict the same outcome. Direction and location has also been described by several labs.

- Overall my reaction to the manuscript is that it would be better suited for a more specialized journal.

Reviewer #3:

In this study, Gmaz and colleagues investigate the responses of single neurons in the rat NAc to reward-predicting cues. The critical finding is that the neurons encode the features of the cues (and do so until and even after the reward occurs). This finding suggests that the responses in NAc may serve as an eligibility trace for learning.

The key finding of the manuscript is that NAc carries cue-specific information. In neuroeconomic terms, it does not use a common currency system to encode values of offers. Instead its neurons link specific offers to their values. This linkage would appear to have implications for neuroeconomic theories of VS function, especially those that see it as a site of the common currency signal. I think the authors are missing an opportunity to highlight an important element of their finding above and beyond the ones they do highlight.

One of the key analyses in the paper concerns using a GLM to determine which units encode which variables and how those relate to each other. The conclusion is that the sets are separate but overlapping. While there is nothing wrong with this analysis, it should be straightforward to do a more sensitive one: correlate the unsigned (absolute value) coding coefficients. This throws away less information and is therefore able to address the question of whether the overlap is precisely what would expect by chance (no correlation), whether it is less than chance (negative correlation, and a bias towards separate populations), or greater than chance (positive correlation, and bias towards a single population).

---

## [Author Response]

Essential revisions:The reviews were overall positive, but two reviewers had significant trouble parsing the rich data presentation to see the key analysis. Thus the truly essential revision is to highlight the critical comparisons that show globally how the encoding of features is dissociable from the encoding of reward. Is this information truly separate in the population, merely orthogonal? Embedded in the reward coding? This can be done with the current single unit analysis, but an interesting adjunct might be to analyze the neurons as populations and remove variance related to the different variables. This or similar analyses would be a nice addition to allow a clear demonstration of what information is in the activity if the authors think it is possible. Alternatively simply clarifying the current approach, maybe at the reduction of some of the anecdotal description might do it.

We are grateful for this observation, which prompted us to create this streamlined revision which we believe is much more clear. Most importantly, we have changed the conceptual schematic to include explicit definitions (coding in *separate* populations, *independent* populations, or in a *joint* population; Figure 1) and associated specific analysis criteria. These definitions are now used throughout the text, and the associated expected patterns of results are shown alongside the observed results to facilitate comparison (Figure 4C-D, Figure 7C-F). We have also reduced the number of figures in the main text to 8.

In addition to this, there are some potential confounds raised that the authors should address. The problem with response direction is important and also drift with or across block and how this is handled should be made clearer.We apologize for not having made our strategy to address this sufficiently clear in the original manuscript. We use a three-pronged approach: *first*, units that show an obvious change in firing rate within a block (significant difference between the first and second half within a block) are excluded from further analysis. *Second*, all our main analyses are performed with stepwise generalized linear models (GLMs) that include time as a nuisance regressor, such that variability (linearly) attributable to time is regressed out. *Finally*, we perform an explicit comparison of variability within blocks versus across blocks, and show that the across-block variability is greater. Thus, any remaining differences between blocks are best explained by differences related to the cues present (light vs. sound). Please see our point-by-point reply below for our responses to other specific concerns.Reviewer #1:This paper reports on neural correlates in rat NAc during performance of a square maze task in which auditory or visual cues delivered in different blocks of trials signaled reward or non-reward down alleys at each corner of the square. […] Actually my main criticism is probably that this makes the main point difficult to maintain and keep track of. I felt like there was too much description of the various anecdotal results in the text and figures and not enough clear demonstration of the independence or orthogonal nature of the cue representations from value. I have a couple of suggestions for this, but I'm sure the authors will also have ideas.For me the main point to make clear is that the neural activity contains information about the cues, contexts, location etc. that is not driven by reward. Note this could be independent, orthogonal, or even integrated. Currently the authors show this now with their GLM analysis of individual unit firing and by comparing proportions of neurons doing different things and amount of overlap by Chi-Squared. I think this is fine, however I think it leads to some complexity and confusion. For example, they say "Nac encodes features of reward-predicting cues separate from expected outcome" referencing Figure 4A among others. While this neuron might show this, it might also (I think) be reflecting the conjunction of sound and reward. Which do they mean? Then just after this they say "overlap of coding of cue features within units was not different than expected by chance according to Chi-Squared tests, suggesting for integrated coding across various aspects of a cue". It seems to me that this shows a lack of integrated coding if it is no more than chance? I was left a bit unsure about how independent representations were.

Thank you for highlighting that the analyses that most directly speak to the major focus of the study were not sufficiently clear. A similar point was raised by reviewer #3. In response, we have made the following changes:

- We have changed the conceptual schematic (Figure 1) to establish the key terminology to describe the scenarios tested. These are coding (for cue outcome and cue identity) in (a) separate populations or (b) overlapping populations, where (b) is further broken down into independent or joint coding. Importantly, these different terms are assigned explicit analysis criteria: coding in separate populations implies a negative correlation coefficient between regression coefficients, independent coding implies zero correlation, and joint coding implies a positive correlation. We define these terms and accompanying criteria in the Introduction and Figure 1 legend, and then consistently use them throughout the text.

- Following a suggestion by reviewer #3, we now test the different neural coding hypotheses by recoding beta coefficients for a cue feature (regressor) to a 1 if the cue feature was a significant predictor in the final model, and 0 if it was not, and calculated Pearson’s *r* comparing an array of the coded 0s and 1s for cue-modulated units for one cue feature with a similar array for another cue feature. This analysis is described in detail in the Materials and methods (subsection “Neural data”, second paragraph).

I wonder if the authors can be much more explicit with the Venn diagrams and comparisons about what is above chance for the unit analysis. Then I think it would be useful to apply an analysis across units – using pseudoensembles – to ask what information is present with and without value present. This would allow the authors to demonstrate clearly that the units, either individually or as a population, contain information about features that is orthogonal to the reward. A similar strategy could be used to show whether the feature information is also integrated or not I think.

We have clarified in the text (subsection “Neural data”) how we determine whether a given unit encodes a given variable (cue identity, outcome, location). In short, this is done through model comparison of different GLMs, with and without the feature of interest. We have also z-scored the observed proportion of units that code for a given variable relative to a shuffled distribution to quantify when our data deviated from chance. Subsequent analyses (correlations on GLM coefficients) and visualizations (correlation matrices) use the output of this important initial step. As above, a correlation (on GLM coefficients) of zero is consistent with independent coding, i.e. overlap expected by chance, and deviations from zero are consistent with separate populations (anticorrelated) or joint coding (positively correlated). Additionally, we have replaced the Venn diagrams with correlation matrices (Figure 4C-D and Figure 7C-F) that summarize the many numbers in a single Venn diagram into a single correlation measure, as well capture the temporal dynamics of cue feature coding throughout a trial.

Regarding the reviewer’s comment on “what information is present with and without value present”, we believe this is addressed naturally by our GLM approach, which asks how much additional variance in firing rate is explained by factors of interest after the effects of others are accounted for. Additionally, we implemented the suggestion to use a pseudoensemble-based analysis, training a classifier to predict various cue feature conditions, and have included alongside the single-unit analysis in Figure 4. These additions are now explained in the subsection “Data analysis: Neural data” in the Materials and methods section.

Reviewer #2:[…] I think the task is overly complicated to answer this question. There are too many confounds and it is not clear when animals hear/see stimuli. Auditory and visual stimuli are presented in different blocks, which leads to confounds related to context, current strategy (authors admit this one) and drift. There should have been a third block during which the modality of the cue was the same as the first block to show genuine cue selectivity. Over, it was not clear to me why auditory and visual stimuli were not used in the same block of trials. Such a task design would have been far more convincing.

We have attempted to clarify the location of cues in the task schematic. Also, while there may have been a perceptual delay between onset of the stimuli and perception by the rat, we believe that this does not affect our primary finding that coding for cue identity is present in the NAc.

As we explained in our reply to reviewer #1’s comments above, we address the possible confound of slow changes across the recording session creating an artificial cue identity signal in three complementary ways:

- First, we exclude units from our GLM analysis that show substantial variability in cue-evoked firing rates between the first and second half of trials within a block, as measured by a Mann-Whitney U test.

- Second, in the GLM, we include trial number as a nuisance variable to remove firing rate variance attributed to the passage of time.

- Third, we compare tiling across the first and second half of a block for a cue condition as a control comparison against tilling across cue conditions.

We note that these approaches are in line with what is commonly done in other studies using blocks with different reward contingencies (e.g. Tryon et al. Hippocampus 2017; Mankin et al. Neuron 2015; Kraus et al. Neuron 2013).

We have expanded our discussion of the limitations of our task to include why we did not have data from sessions where both sets of cue were presented in the same block of trials (Discussion subsection “Identity coding”, last paragraph). We piloted having the animals being presented with both sets of stimuli in the same block and having them switch between stimulus dimensions, but they would not learn to switch strategies within a reasonable timeframe, even after several switches.

- Another confound that resulted from task design is that rewards are only provided for left turns. Right turns only led to the next decision point. Thus neurons that appear to be encoding left and right might be signaling no reward or the possibility of reward.

The main objective of our study was to test to what extent NAc neurons encode cue identity. For completeness, we also sought to determine whether coding of cue identity overlapped with coding of other task variables (specifically, cue outcome and location). The reviewer is correct that left and right turns may correlate with cue outcome, which means that a neuron that we classified as encoding cue outcome may actually be encoding left vs. right turn. We controlled for this possibility by including left/right turn in the stepwise GLM procedure and only counting a neuron as encoding cue outcome if adding cue outcome improved the model. Thus, left/right selectivity cannot explain these results. We have clarified this in the Results (subsection “NAc encodes behaviorally relevant and irrelevant cue features”, second paragraph).

Additionally, we ran our cue-onset GLM only on those trials where rats turned left, making their behavior similar for both reward-available and reward-unavailable trials. The GLM analysis revealed a similar pattern of results for outcome coding (20% vs. 26% of cue-modulated units for the original and approach only GLM, respectively; see the aforementioned paragraph), further suggesting that units were encoding expected outcome and not subsequent behavior.

- I did not find the plotted histograms overly informative. Many of them simply plot firing rates associated with the trial-types that the neurons were selected for. It would be more useful to show us those trials that the neurons were selective for but also break trials down in the other parameters. For example, for neurons that prefer light blocks over sound blocks (or vice versa), we should be shown those trial-types but also trials broken down by the decision the animal made and if reward was available. This strategy should be done for cells that are selective for block, reward, and arm, at the single neuron and population level.

Thank you for this suggestion. We felt that including all these plots in the original figure made it unwieldy, so we decided to include expanded plots for all cue features and trial types as supplements to the main example figures (Figures 3 and 6).

- I'm not convinced that what we are seeing in single neuron and population histograms is not due to drift. Too much of the selectivity is observed before cue onset. This is a problem for all cell types but even a bigger problem for neurons selective for reward availability.

Thank you for this insightful comment. As we see it, there are two distinct issues here:

- First, an important “sanity check” is to verify that before cue onset, there is no coding of cue features that have not yet been presented. Specifically, it should not be possible for the rat to predict cue outcome (rewarded or not) beyond the pseudorandom nature of the trial order.

To check this, we calculated the z-score for each cue feature proportion at each time point, comparing that value to the mean and variance of our shuffled samples, and added a marker indicating when this observed value deviated from the shuffled samples, measured by a z-score > 1.96. This quantification revealed that while both cue identity and cue location were encoded by a proportion of units significantly higher than the shuffled distribution at this pre-cue time epoch, cue outcome was not, suggesting perhaps context coding (cue identity) in the NAc, but not for the outcome of the cue itself (Figure 4A and accompanying text, subsection “NAc encodes behaviorally relevant and irrelevant cue features”, second paragraph).

Additionally, our task design was pseudorandom such that the same cue could not be present for more than two trials in a row, making our task somewhat predictable to the astute rat. As we did in the original submission, we include task history as a regressor in our stepwise GLM, such that firing rate variance due to previously presented cues is excluded. We explain this in the Materials and methods (subsection “Neural data”).

- The second issue is the question, is cue identity coded only following the cue, or more generally throughout a trial and block? Our analysis (Figure 7) suggests both occur; either way, our results show that vStr neurons encode much more than reward-related variables only. We have expanded our discussion of this issue in the identity coding section (Discussion, first paragraph).

- Along this same issue there should be figures that show how selectivity emerged in both trial blocks.

We agree that this is an interesting question. However, our study was designed to assess performance during a well-learned behavior, and as a result we did not observe learning within blocks. We have incorporated a statement in the behavioral results that points out the lack of differences in behavioral performance within a block (Results, subsection “Behaviour”). Thus, an examination of how selectivity emerges would require a different experimental design.

- Overall figures are too small making it very difficult to see things. For example the differently colored rasters are impossible to see. They should be bigger. Also, the information carried in those rasters related to trial-types should be shown in the average line plots below as suggested above.

We made the figures larger where we could, and have added the PETHs for all the cues presented in the expanded plots (see supplements for Figures 3 and 6).

- Percentage should be provided in the table. Also, some indication if counts are significantly more than chance should be added to the table.

We have added the percent of units (relative to the total of 133 cue-modulated units) influenced by a given predictor to Table 1.

Regarding chance levels for the reported cell counts, there are two analyses that use these counts, and each has a corresponding chance level determined by shuffles.

For characterizing the percentages (Table 1), we used a conservative criterion in the stepwise GLM to determine whether or not a predictor should be added to the model (improve deviance with a p < 0.01 to be included). Indications of whether counts are significantly more than chance have been added to Figure 4A (subsection “NAc encodes behaviorally relevant and irrelevant cue features”, second paragraph). As we explain in response to an earlier point, we determined whether or not the observed count for a cue feature at a given time point was significantly more than chance by comparing to the mean and standard deviation of a shuffled distribution, generated by shuffling the order of firing rates for a unit (subsection “Neural data”).

In terms of counts of overlap among cue features, we have implemented a variant of a suggestion of reviewer #3 where we created an array of 0s and 1s for all units of a cue feature, with 0 representing a unit where the cue feature wasn’t a predictor in the model, and 1 representing a unit where the cue feature was a predictor in the model. We correlated this array of 0s and 1s across cue features to see if units with overlapping counts was different from chance, compared to a shuffled distribution (see the aforementioned subsection).

- The point of the results illustrated in Figure 7 and 11 is not clear. Scatter plots (where each dot is a neuron), that plot firing on light blocks versus sound blocks would better show if neurons are or are not encoding expected reward across modalities.

We apologize for not making this sufficiently clear in the original submission. The purpose of Figures 7 and 11 in the original submission (now Figure 5 and a supplement to Figure 7) is to illustrate that a) the NAc encodes all dimensions of task space and b) it does so in a way that covers the entire duration of a trial, not only time-locked to the reward-predictive cue. As such, we believe these figures provide a powerful visualization of our overall conclusion that NAc provides much richer signals than merely a reward-related response to a cue and reward itself. We have expanded the discussion of this point in the third paragraph of the subsection “Identity coding”.

Additionally, we have added the requested scatterplot of light block firing vs. sound block firing as Figure 4—figure supplement 2.

- It is not always clear what trial types are being analyzed throughout the paper; correct trials only?

All available trials were used for the analyses, which were all trials in a session for the cue-onset analyses, and all approach trials in a session for the nosepoke analyses. The following four excerpts indicate the text where this information is present:

Results:

“Fitting GLMs to all trials within a session revealed that a variety of task parameters accounted for a significant portion of firing rate variance in NAc cue-modulated units (Figure 4A, Figure 4—figure supplement 1, Figure 4—figure supplement 2, Table 1).”

“To quantitatively test whether representations of cue features persisted post-approach until the outcome was revealed, we fit sliding window GLMs to the post-approach firing rates of cue-modulated units aligned to both the time of nosepoke into the reward receptacle, and after the outcome was revealed (Figure 6A, B, Figure 6—figure supplement 1A-D, Table 1).”

Methods:

“To determine the relative contribution of different task parameters to firing rate variance (as in Figure 4A, Figure 4—figure supplement 1), a forward selection stepwise GLM using a Poisson distribution for the response variable was fit to each cue-modulated unit, using data from every trial in a session.”

“To identify the responsivity of units to different cue features at the time of nosepoke into a reward receptacle, and subsequent reward delivery, the same cue-modulated units from the cue-onset analyses were analyzed at the time of nosepoke and outcome receipt using identical analysis techniques for all approach trials (Figures 5, 6).”- Liner models are used but is firing normal?

Firing is not normal. We used generalized linear models, which have the added benefit over ordinary linear regression of allowing the selection of a non-normal distribution for the response variable. We used a Poisson distribution in our GLMs to model the spiking data, as the Poisson distribution only has positive integer values and using a Poisson distribution to model spiking rates is the current standard in the field (albeit we recognize its shortcomings; that is, the biologically unrealistic assumption of independence of ISIs and lack of refractory period; furthermore, may not be the best model for all neurons, see Maimon and Assad 2009).

Furthermore, we recognize the error in the data analysis section of the Materials and methods section where we said we used ‘general linear models’ where the response variable follows a normal distribution instead of the more flexible ‘generalized linear models’ which is what was actually used. This typo has been corrected.

- These neurons might be involved in credit assignment but I think the authors' claims are too far reaching. There is no inactivation. There is no link between firing and behavior, and rats are already well trained on the task.

We agree, and merely wish to point out that representation of cue identity is a prerequisite for temporal credit assignment. Showing that the signal reported here is in fact used for credit assignment requires a line of research that performs perturbations at specific key learning moments, performs reinforcement learning model fits with strength of the signal as a regressor, and other experiments. Showing this signal exists we believe is a useful first step and contribution to the literature that will encourage future work directed at these important issues, as well as provide a counterpoint to the dominant narrative that NAc neurons code value. We have expanded the Discussion to further highlight this limitation:

“An overall limitation of the current study is that rats were never presented with both sets of cues simultaneously, and were not required to switch strategies between multiple sets of cues (this was attempted in behavioral pilots, however animals took several days of training to successfully switch strategies). […] Thus, it is unknown to what extent the cue identity encoding we observed is behaviorally relevant, although extrapolating data from other work (Sleezer et al., 2016) suggests that cue identity coding would be modulated by relevance.”

- How long were cues on?

We have made this explicit in the Materials and methods (subsection “Behavioral task and training”, first paragraph). Specifically, cues were on from when the rat triggered a track photobeam until 1 second after outcome receipt on approach trials, and until initiating the following trial on skip trials.

- Early work by Schultz examining these issues should be mentioned.

Thank you for this suggestion; we have added the 1992, 2001, and 2003 Schultz papers to the reward coding references in the Introduction.

- These effects are not that novel, as the authors point out, because others have already shown odor selectivity. This is true even in reversal tasks where the different cues predict the same outcome. Direction and location has also been described by several labs.

We apologize for not having made a crucial difference with prior work sufficiently clear. A novel contribution of our paper is that we report distinct NAc coding for two sets of equally predictive cues. In contrast, the Setlow et al.. 2003 study reported that NAc units tracked the motivational significance (outcome) of their odor cues. Thus, following reversal of the cue-outcome relationship, NAc units switched their selectivity from the previously outcome-predictive cue for that outcome to the currently outcome-predictive cue. We have clarified this further in the following excerpt from the Discussion:

“Setlow et al., 2003 paired two distinct odor cues with appetitive and aversive odor cues respectively in a Go/NoGo task, such that cue identity and cue outcome were linked. […] Our results suggest that the NAc dissociates cue identity representations at multiple levels of analysis (e.g. single-unit and population) even when the motivational significance of these stimuli is identical.”

- Overall my reaction to the manuscript is that it would be better suited for a more specialized journal.Reviewer #3:[…] The key finding of the manuscript is that NAc carries cue-specific information. In neuroeconomic terms, it does not use a common currency system to encode values of offers. Instead its neurons link specific offers to their values. This linkage would appear to have implications for neuroeconomic theories of VS function, especially those that see it as a site of the common currency signal. I think the authors are missing an opportunity to highlight an important element of their finding above and beyond the ones they do highlight.

We are grateful for the idea to expand the conceptual framework and the opportunity to reach a broader audience. We have added the following to the Introduction and Discussion sections, respectively:

“Similarly, in a neuroeconomic framework, NAc is thought to represent a domain-general subjective value signal for different offers (Peters and Büchel, 2009; Levy and Glimcher, 2012; Bartra et al., 2013; Sescousse et al., 2015); having a representation of the offer itself alongside this value signal would provide a potential neural substrate for updating offer value.”

“Viewed within the neuroeconomic framework of decision making, functional magnetic resonance imaging (fMRI) studies have found support for NAc representations of *offer value*, a domain-general common currency signal that enables comparison of different attributes such as reward identity, effort, and temporal proximity (Peters and Büchel, 2009; Levy and Glimcher, 2012; Bartra et al., 2013; Sescousse et al., 2015). Our study adds to a growing body of electrophysiological research that suggests the view of the NAc as a value center, while informative and capturing major aspects of NAc processing, neglects additional contributions of NAc to learning and decision making such as the offer (cue) identity signal reported here.”

One of the key analyses in the paper concerns using a GLM to determine which units encode which variables and how those relate to each other. The conclusion is that the sets are separate but overlapping. While there is nothing wrong with this analysis, it should be straightforward to do a more sensitive one: correlate the unsigned (absolute value) coding coefficients. This throws away less information and is therefore able to address the question of whether the overlap is precisely what would expect by chance (no correlation), whether it is less than chance (negative correlation, and a bias towards separate populations), or greater than chance (positive correlation, and bias towards a single population).

Thank you for this insightful and valuable suggestion, which has enabled us to make the manuscript much more clear and streamlined. As we explain in the schematic overview (Figure 1) this new analysis attaches specific correlation values to the hypothesized coding schemes (separate coding, independent coding, joint coding), which we reference throughout the text. As we explain in the Materials and methods and use in the Results, respectively:

“To determine the degree to which coding of cue identity, cue location, and cue outcome overlapped within units we correlated the recoded beta coefficients from the GLMs for the cue features (Figure 4C, D). […] To summarize the correlation matrices generated from this analysis, we put the output of our 100 shuffled GLMs through the same pipeline, took the mean of the 36 correlations for a block comparison for each of the 100 shuffles for an analysis window, and used the mean and standard deviation of these shuffled correlation averages to compare to the mean of the comparison block for the actual data.”